# Divergence of TORC1-mediated stress response leads to novel acquired stress resistance in a pathogenic yeast

Jinye Liang[1]*, Hanxi Tang[1]☿, Lindsey F. Snyder[2]☿, Christopher E. Youngstrom[1], Bin Z. He ᴵᴰ[1,2]*

1 Department of Biology, The University of Iowa, Iowa City, Iowa, United States of America,
2 Interdisciplinary Graduate Program in Genetics, The University of Iowa, Iowa City, Iowa, United States of America

☿ These authors contributed equally to this work.
* jinye-liang@uiowa.edu (JL); bin-he@uiowa.edu (BZH)

## Abstract

Acquired stress resistance (ASR) enables organisms to prepare for environmental changes that occur after an initial stressor. However, the genetic basis for ASR and how the underlying network evolved remain poorly understood. In this study, we discovered that a short phosphate starvation induces oxidative stress response (OSR) genes in the pathogenic yeast *C. glabrata* and protects it against a severe $H_2O_2$ stress; the same treatment, however, provides little benefit in the low pathogenic-potential relative, *S. cerevisiae*. This ASR involves the same transcription factors (TFs) as the OSR, but with different combinatorial logics. We show that Target-of-Rapamycin Complex 1 (TORC1) is differentially inhibited by phosphate starvation in the two species and contributes to the ASR via its proximal effector, Sch9. Therefore, evolution of the phosphate starvation-induced ASR involves the rewiring of TORC1's response to phosphate limitation and the repurposing of TF-target gene networks for the OSR using new regulatory logics.

**Data Availability Statement:** All data for this submission can be accessed at https://doi.org/10.5281/zenodo.10004973, or are included in the supplemental information, with the exception of the

## Author summary

Acquired Stress Resistance (ASR) is a phenomenon where mild stress makes an organism more resilient to subsequent severe stress. In this study, we uncovered a unique ASR in the opportunistic yeast pathogen *C. glabrata* compared to its less pathogenic relative *S. cerevisiae*. When subjected to a non-lethal phosphate starvation, *C. glabrata* activates genes that enhance its resistance to severe $H_2O_2$ stress, making it survive 3–10 times better than naïve cells, while the same treatment offers little to no protection in *S. cerevisiae*. We found that the underlying gene network for ASR shares key components with the typical oxidative stress response, but operates with different regulatory logics. Notably, the Target-of-Rapamycin Complex 1 (TORC1) responds differently to phosphate limitation in the two species, underpinning the species divergence in ASR. This discovery highlights a species-specific ASR and the key genetic factor driving the divergence. These findings shed light on how pathogenic yeasts adapt to their host environments.

RNA-seq data for phosphate starvation in C. glabrata, which is available through the GEO database at https://www.ncbi.nlm.nih.gov/geo/query/acc.cgi?acc=GSE244380.

**Funding:** BZH is supported by NIH R35GM137831 and a start-up fund provided by the University of Iowa. The funders had no role in study design, data collection and analysis, decision to publish, or preparation of the manuscript.

**Competing interests:** The authors have declared that no competing interests exist.

## Introduction

Acquired stress resistance (ASR) is the phenomenon where organisms exposed to a non-lethal primary stress become more resistant to a secondary severe stress of the same or a different type [1–6]. Widely observed across kingdoms of life, ASR is believed to provide a survival advantage in environments with predictable changes. For example, it was shown in *S. cerevisiae* that stressors occurring early during fermentation, e.g., heat and ethanol, have a strong protective effect against stresses that occur later, such as oxidative stress, while the reverse is not true [7]. This asymmetry in cross-stress protection suggests that ASR is not simply a general stress response but rather represents an adaptive anticipatory response, enabling cells to predict and survive future challenges. A natural prediction is that ASR must be adapted to the specific stress patterns organisms encounter in their environments, leading to divergence between species.

Despite the potential role of ASR in adaptation, we know little about its evolution. ASR is known to have biochemical, post-translational or transcriptional mechanisms [8,9]. At the transcriptional level, studies in *S. cerevisiae* showed that ASR does not rely on a single, all-purpose program; instead, genes involved in ASR are regulated in a stress-specific manner [10]. Furthermore, ASR against the same secondary stress is dependent on distinct genes that are regulated by different types of primary stresses [11]. Such stress-specific regulation enables mutations to target specific contexts and avoid pleiotropic effects, thereby facilitating the evolution of ASR. However, few studies dissected ASR in closely related species. Previous studies on ASR in yeasts, for example, have primarily focused on comparing *S. cerevisiae* with distantly related species, such as *C. albicans* and *S. pombe*, which are estimated to have diverged from the baker's yeast ~200 and ~500 million years ago [12–16]. These substantial evolutionary distances, coupled with the whole genome duplication (WGD) event specific to *S. cerevisiae*, significantly hindered our ability to discern the evolutionary alterations behind ASR divergence.

In this study, we focused on the more closely related *S. cerevisiae* and *C. glabrata*, both of which are post-WGD and share more than 90% of their genes with one-to-one orthologs [17]. Even though *C. glabrata* is evolutionarily close to *S. cerevisiae*, it is distinguished by its ability to colonize and infect humans [18,19]. As one of the most frequent causes for Candidiasis, it shares many phenotypic traits with the distantly related pathogen, *C. albicans*, including high resistance to host-associated stresses such as the oxidative stress [20,21]. In fact, it exhibits the highest resistance to H₂O₂ in a broad phylogenetic survey of fungi [22]. Apart from oxidative stress, host immune systems also use other stressors, such as nutrient limitation, to kill invading microbes [23]. Given the array of challenges they encounter, it is probable that yeast pathogens such as *C. glabrata* evolved anticipatory responses to help them prepare for impending stresses [16]. Interestingly, we and others found that *C. glabrata*, *C. albicans* and a more distantly related pathogen, *Cryptococcus neoformans*, all have an expanded phosphate starvation (PHO) response compared with that in *S. cerevisiae*, including the induction of oxidative stress response (OSR) genes [24–26]. As an essential macronutrient, fungal cells must acquire phosphate from the environment. Emerging evidence suggests that responses to phosphate limitation play a crucial role in the host-fungal interaction. First, a defective PHO pathway, either by deleting the transcription factor (TF) *PHO4* or the high affinity phosphate transporter *PHO84*, leads to reduced virulence and survival for *C. albicans* inside macrophages and animal models [24,27–29]. Second, two clinical isolates of *C. albicans* exhibit enhanced virulence in a phosphate-dependent manner in both a *C. elegans* and a mouse model of infection [30]. Third, several PHO genes were found to be upregulated in phagocytosed *C. glabrata* cells, suggesting that yeast cells experience phosphate limitation inside the phagosome [31].

We found in this study that a short-term phosphate starvation led to the induction of at least 15 OSR genes and enhanced the survival of *C. glabrata* against a severe $H_2O_2$ stress. The same treatment had little to no effect in the related *S. cerevisiae*. Using transcriptional profiling, genetic perturbations, reporter and survival assays, we identified key players in the network underlying the ASR, including the catalase gene *CTA1*, transcription factors (TFs) Msn4 and Skn7 and homolog of the Greatwall kinase, Rim15. We found that phosphate limitation quickly and strongly suppressed the Target-of-Rapamycin Complex 1 (TORC1) in *C. glabrata* but not in *S. cerevisiae*; the proximal effector Sch9 was also shown to contribute to the ASR in the former species, strongly implicating TORC1 as a major contributor to the divergence in ASR. In summary, our results identified a key signaling-TF-effector gene subnetwork underlying the divergent ASR phenotype and thereby provides a concrete example of ASR evolution.

## Results

### Phosphate starvation provides strong acquired resistance for $H_2O_2$ in *C. glabrata* but not in *S. cerevisiae*

Based on the evidence linking phosphate to oxidative stress and virulence in *C. glabrata* and *C. albicans*, we asked whether phosphate starvation could provide acquired resistance for severe $H_2O_2$ stress in *C. glabrata* and whether the behavior is different in the low pathogenic-potential relative, *S. cerevisiae*. To answer this question, wild type cells from both species were subjected to 45 minutes of phosphate starvation, followed by a severe $H_2O_2$ challenge (Fig 1A). We designed the experiment to be able to compare the ASR effect between species despite differences in their basal resistance level to $H_2O_2$, which affects the magnitude of ASR [32]. Through titration, we found that 100 mM and 10mM $H_2O_2$ resulted in a similar ~2.5% survival in *C. glabrata* and *S. cerevisiae*, respectively (S1A Fig, *P* = 1, Materials and Methods). Using these concentrations as the secondary stress, we found that phosphate starvation greatly enhanced the survival of *C. glabrata* after the $H_2O_2$ stress (Fig 1B) but had no detectable effect in *S. cerevisiae* (Fig 1C). A quantitative colony forming unit (CFU) assay confirmed this (Fig 1D, raw *P* = 0.0039 and 0.37 in *C. glabrata* and *S. cerevisiae*, respectively). To quantify the magnitude of ASR, we defined ASR-score as the fold increase in survival due to the primary stress treatment, i.e., the ratio between the survival rates with (r') and without (r) the primary stress. Using this definition, we found a 45-minute phosphate starvation led to an ASR-score of 3.4 in *C. glabrata* (95% CI: [2.66, 4.26]), compared with 1.3 in *S. cerevisiae* (95% CI: [0.90, 1.63]). To comprehensively test the relationship between phosphate starvation and ASR for oxidative stress, we altered both the severity of the secondary stress and the length of the primary stress and asked how this affected ASR in both species. We found that more severe secondary stress (higher [$H_2O_2$]) led to stronger ASR in *C. glabrata* but not in *S. cerevisiae* at the concentrations tested (S1B Fig). When we increased the primary phosphate starvation treatment to 90 and 135 minutes, we observed a stronger ASR in *C. glabrata* (ASR-scores = 111 and 170, *P* < 0.05 after Bonferroni correction, S1C Fig). A similar trend was observed in *S. cerevisiae* (ASR-score = 11 and 22) but didn't reach statistical significance (Bonferroni-corrected *P* = 0.7 and 0.5).

We also wondered whether the ASR for $H_2O_2$ is systematically different between the two species or whether phosphate starvation represents a distinct case of divergence. To answer this question, we tested two other stresses, heat shock and glucose starvation, and found that both led to ASR at a similar level between the two species (S2 Fig). Therefore, we conclude that *C. glabrata* and *S. cerevisiae* diverged particularly in the phosphate starvation-induced ASR for $H_2O_2$, where *C. glabrata* showed a strong ASR while the same primary stress elicited a much weaker effect in *S. cerevisiae*.

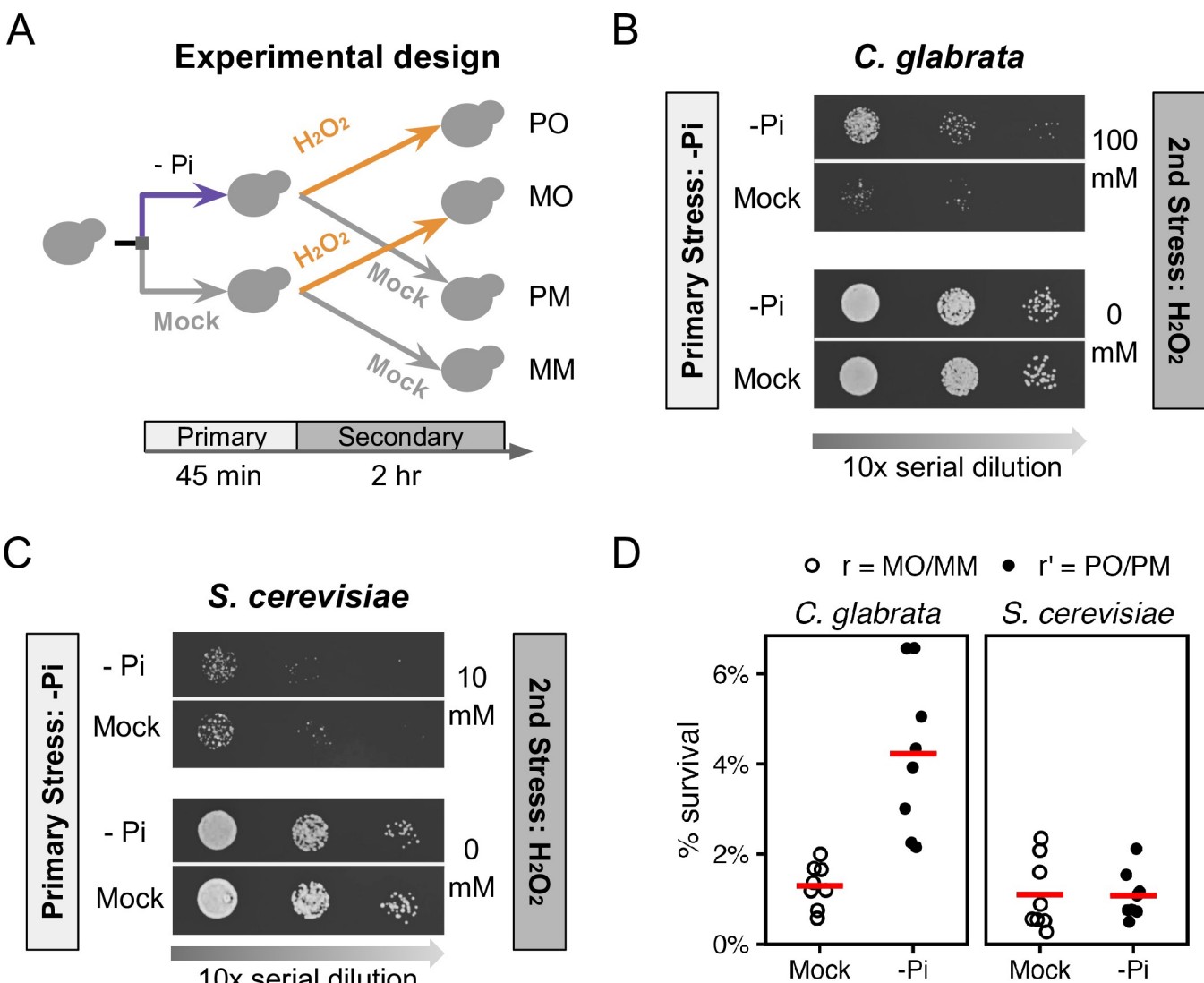

**Fig 1. Phosphate starvation induces strong acquired resistance for H$_2$O$_2$ in *C. glabrata* and not in *S. cerevisiae*.** (A) Experimental design for quantifying Acquired Stress Resistance (ASR): cells were first treated with phosphate starvation (-Pi, or P) or mock-treated (M), then exposed to a secondary H$_2$O$_2$ (O) stress or a mock treatment (M). The resulting treatment regimens were referred to as PO, MO, PM and MM. After each treatment regimen, *C. glabrata* (B) and *S. cerevisiae* (C) cells were spotted on rich solid media and imaged after 24–48 hrs. Different H$_2$O$_2$ concentrations were used for the two species to achieve a similar survival rate without the primary stress. (D) Survival rates (%) after the secondary H$_2$O$_2$ challenge were quantified using Colony Forming Units either with (r') or without (r) phosphate starvation as a primary stress. The biological replicates (n = 8) were shown as dots and their means as bars. Basal survival rates (r) were not different between species (*P* = 0.38), while there was a significant increase in survival due to the primary stress in *C. glabrata* but not in *S. cerevisiae* (*P*-values = 0.0039 and 0.37, respectively).

Lastly, we asked if phosphate starvation provides protection for other oxidative stress agents by replacing the secondary stress with either tert-butyl hydroperoxide (tBOOH), an alkyl hydroperoxide, or menadione (menadione sodium bisulfite, MSB), a superoxide agent. We found that phosphate starvation does not provide ASR for tBOOH, while a moderate ASR effect was observed for menadione compared to H$_2$O$_2$ (S3 Fig). We conclude that the existence and magnitude of phosphate starvation-induced ASR depends on the nature of the ROS. This result supports the prediction that ASR is specific to the primary and secondary stress combinations rather than being a general cross-stress protection mechanism [7].

### Oxidative stress response genes were induced under phosphate starvation in *C. glabrata*

One potential mechanism for the observed ASR in *C. glabrata* is that phosphate starvation led to the induction of oxidative stress response (OSR) genes during the primary stress (Fig 2A). To test this, we curated a set of H₂O₂-response genes in the well-studied *S. cerevisiae*, which include antioxidants, proteases, chaperones and other genes to scavenge ROS and mitigate ROS-induced damages, as well as the TFs regulating them [33,34]. Using this reference set (S1 Table), we identified their orthologs in *C. glabrata* and asked if they were induced during phosphate starvation. To do so, we profiled the transcriptional response at 1 hour of phosphate starvation in *C. glabrata* and compared it with the response to the same treatment in *S. cerevisiae* from [35]. Of the 29 genes tested, 15 were significantly induced by at least two-fold in *C. glabrata* (FDR < 0.05), compared with three in *S. cerevisiae* (Fig 2B–2E). These include

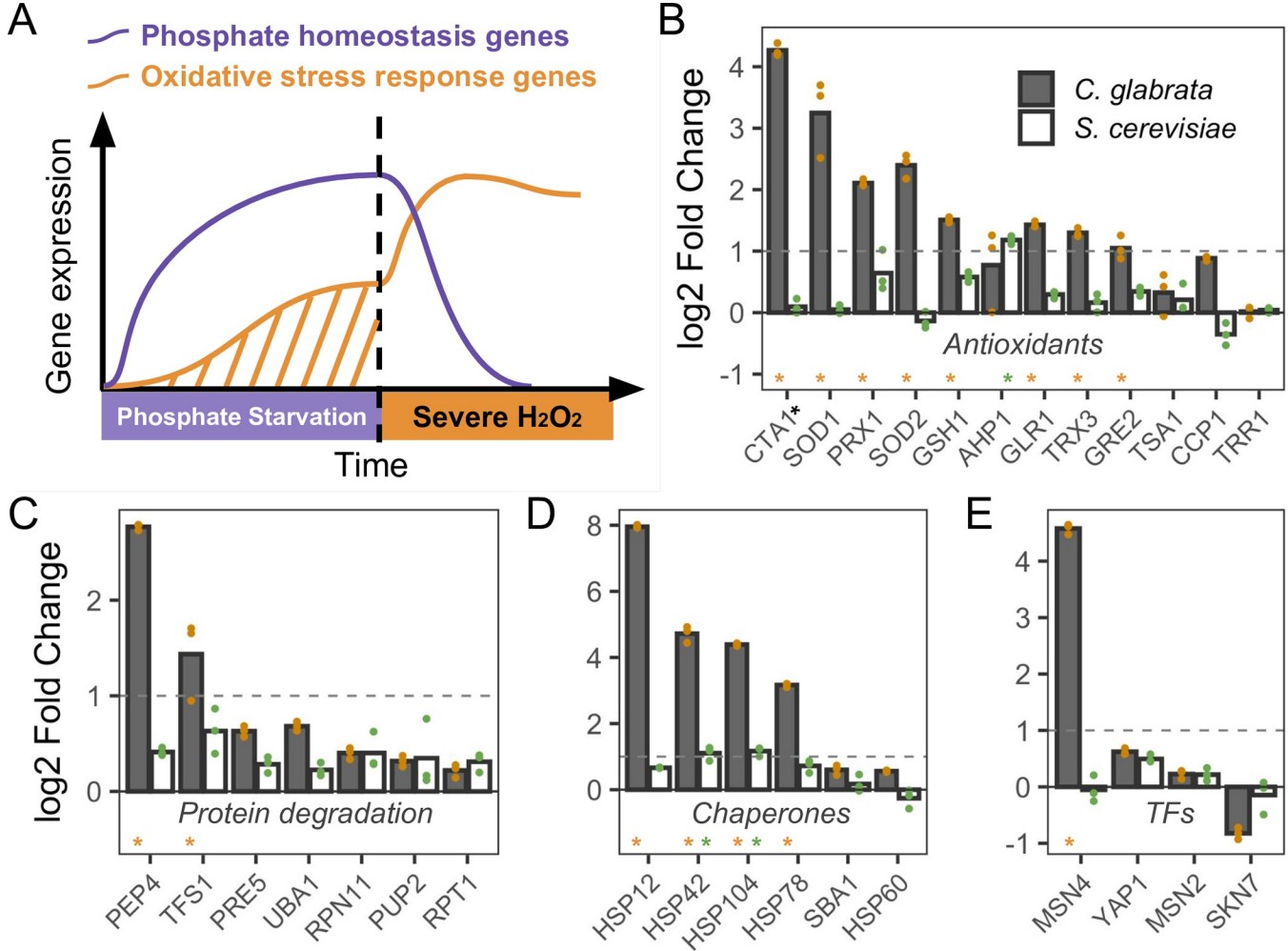

**Fig 2. Phosphate starvation strongly induces many oxidative stress related genes in *C. glabrata* but not in *S. cerevisiae*.** (A) Proposed transcriptional basis for the acquired stress resistance in *C. glabrata*: phosphate starvation induces both canonical phosphate homeostasis genes and also oxidative stress response genes, providing protection for the secondary H₂O₂ challenge. (B-E) Comparison of transcriptional induction of genes known to be involved in OSR in *S. cerevisiae*. Log2 fold changes after 1 hour of phosphate starvation were shown as the mean (bar) of 3 biological replicates (dots) for genes encoding antioxidants (B), protease components (C), molecular chaperones (D) and TFs involved in the OSR (E). The dotted lines indicate a 2 fold induction. Gene names were based on *S. cerevisiae* except for *CTA1*, which had two paralogs in *S. cerevisiae* and only one in *C. glabrata*. *S. cerevisiae CTT1* is involved in OSR and its fold change is shown in (B). Asterisks above a gene name indicate that the gene was significantly induced in that species at an FDR of 0.05.

antioxidant genes such as *CTA1*, *PRX1*, *SOD1* and *SOD2*. Induction of these OSR genes could lead to enhanced ROS scavenging and protection against oxidative damages in *C. glabrata*, consistent with its stronger ASR effect [36–39].

## *CTA1* is required for the ASR for $H_2O_2$ but not during phosphate starvation in *C. glabrata*

To test the hypothesis that the induction of OSR genes contributed to the ASR, we focused on *CTA1*, which has a well-established role in the resistance to $H_2O_2$ in *C. glabrata* [40]. We first used qRT-PCR to confirm that *CTA1* is ~3 fold induced after 45 minutes of phosphate starvation (Fig 3A). We also confirmed that Cta1 protein levels increased during phosphate starvation by endogenously tagging Cta1 with GFP (Fig 3B). Notably, this strain had the same $H_2O_2$ resistance as the wild type (S4A Fig). To determine if *CTA1* is required for the phosphate starvation-induced ASR in *C. glabrata*, we created a *cta1Δ* strain and compared its ASR phenotype to the wild type using comparable strengths of $H_2O_2$. We found that loss of *cta1* largely abolished the phosphate starvation-induced ASR for $H_2O_2$ (Fig 3C). Quantitative CFU assays revealed a residual ASR effect in *cta1Δ* (ASR-score = 1.5, 95% CI [1.2, 1.8]), which was significantly lower than that in the wild type strain (ASR-score = 9.8, 95% CI [5.5, 15], Mann-Whitney's U test comparing the two genotypes, $P$ = 0.001) (Fig 3D). The ASR defect in *cta1Δ* was rescued by putting *CTA1* back into its endogenous locus (S4B and S4C Fig). To determine if *CTA1* induction is sufficient to confer ASR, we replaced the endogenous promoter of *CTA1* with an inducible *MET3* promoter. We confirmed that *MET3pr-CTA1* was induced to a comparable level in the induction media as the endogenous *CTA1* under phosphate starvation at 45 min (S5A Fig). Using this strain, we found that pre-inducing *CTA1* enhanced the survival of the cells during the secondary stress as did phosphate starvation (S5B and S5C Fig). We therefore conclude that *CTA1* induction is both necessary and sufficient for the ASR. It's worth noting this doesn't rule out the contribution of other OSR genes induced under phosphate starvation (Fig 2).

One question that remained was whether the induction of *CTA1* and other OSR genes served strictly to enhance the survival during the secondary stress, or if they were part of the primary stress response, with an unintended benefit for the secondary stress. To distinguish between these two hypotheses, we asked if Cta1, or the main TF for the canonical OSR, Yap1, were required for survival under phosphate starvation. We found that deletion of neither gene affected growth relative to the wild type during phosphate limiting conditions, while a strain deficient in the PHO response showed severe growth defects (S6A Fig). We conclude that the induction of *CTA1* primarily serves the secondary $H_2O_2$ stress. Lastly, we found that while *CTA1* is crucial for the phosphate starvation-induced ASR for $H_2O_2$, it is less important with mild $H_2O_2$ used as the primary stress (S6B Fig). This result supports the previous finding that the genetic basis for ASR is highly variable and dependent on the primary stress [11].

## Msn4 and Skn7 contribute to *CTA1* induction during phosphate starvation in *C. glabrata*

We reasoned that TFs involved in *CTA1* induction during phosphate starvation must be key to the ASR. Four TFs were implicated in *CTA1*'s induction in *C. glabrata* under $H_2O_2$ stress, i.e., Yap1, Skn7, Msn2 and Msn4 [40]. To determine which one(s) contributes to *CTA1* induction under phosphate starvation, we deleted each TF and measured Cta1-GFP induction. We found that *yap1Δ* and *msn2Δ* didn't affect Cta1 induction, while *msn4Δ* and *skn7Δ* both exhibited decreased Cta1 induction during phosphate starvation (Fig 4A). A double mutant *msn4Δ skn7Δ* exhibited a stronger reduction in Cta1-GFP than either TFΔ alone, suggesting that both

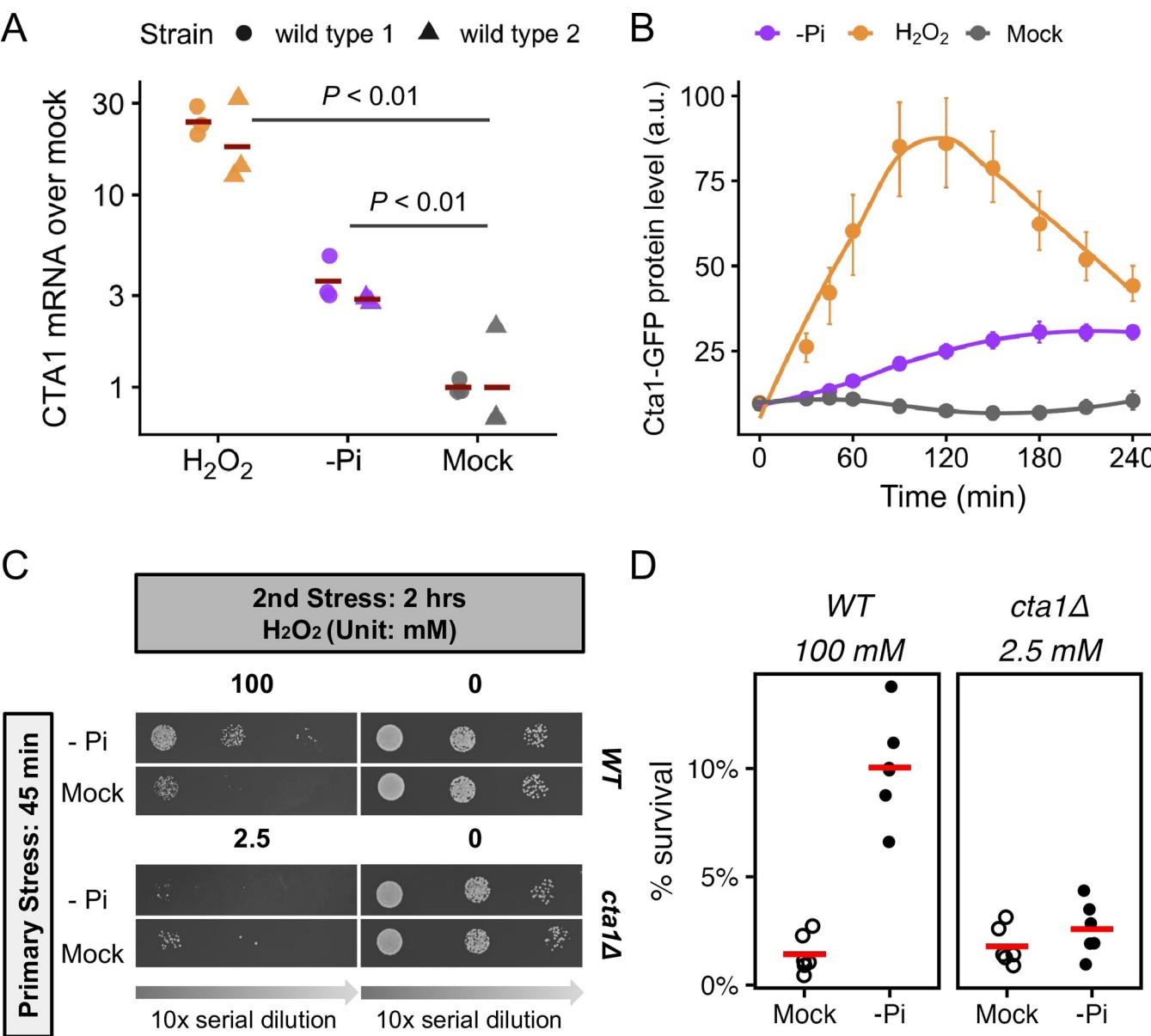

**Fig 3. *C. glabrata CTA1* is induced during phosphate starvation and is required for the acquired resistance to severe $H_2O_2$ stress.** (A) *CTA1* mRNA levels were assayed using qRT-PCR following 45 minutes of either 1.5 mM $H_2O_2$, phosphate starvation or mock treatment. Fold changes over the mock treated samples were shown using *ACT1* as a reference. Two isogenic lab strains of *C. glabrata* were each assayed in triplicates. Bars represent the mean of each strain. An unpaired Student's t-test comparing the $\Delta\Delta C_T$ values between both treatments with the mock found both to be significantly elevated ($P < 0.01$, strain was included as a covariate and found to be not significant). (B) Cta1 protein levels were monitored in strains carrying a genomic *CTA1*-GFP fusion using flow cytometry for 4 hours, during which cells were exposed to either mild $H_2O_2$ stress, phosphate starvation or mock treatment. The y-axis values are the median fluorescence intensities (a.u. = arbitrary unit). The dots show the mean of >3 biological replicates; the error bars show 95% confidence intervals based on bootstrapping and the lines are LOESS curves fitted to the data. (C) ASR spotting assays for wild type and *cta1Δ C. glabrata* strains. 2.5 mM $H_2O_2$ was used as the secondary stress for *cta1Δ*, which resulted in a similar basal survival rate as 100 mM $H_2O_2$ for the wild type. (D) ASR effects were quantified by comparing the survival rates after H2O2 treatments either with (solid circles) or without (open circles) the primary stress using a colony forming unit (CFU) assay. The difference in basal survival rates was not statistically significant between the wild type and the *cta1Δ* strain ($P = 0.39$). Phosphate starvation significantly increased the survival of both strains during the secondary challenge (raw $P = 0.016$ in both), but the ASR effect size was much smaller in cta1Δ (ASR-score = 9.8 and 1.5 in wild type and *cta1Δ*, respectively; Mann-Whitney U test $P = 0.001$ between the two).

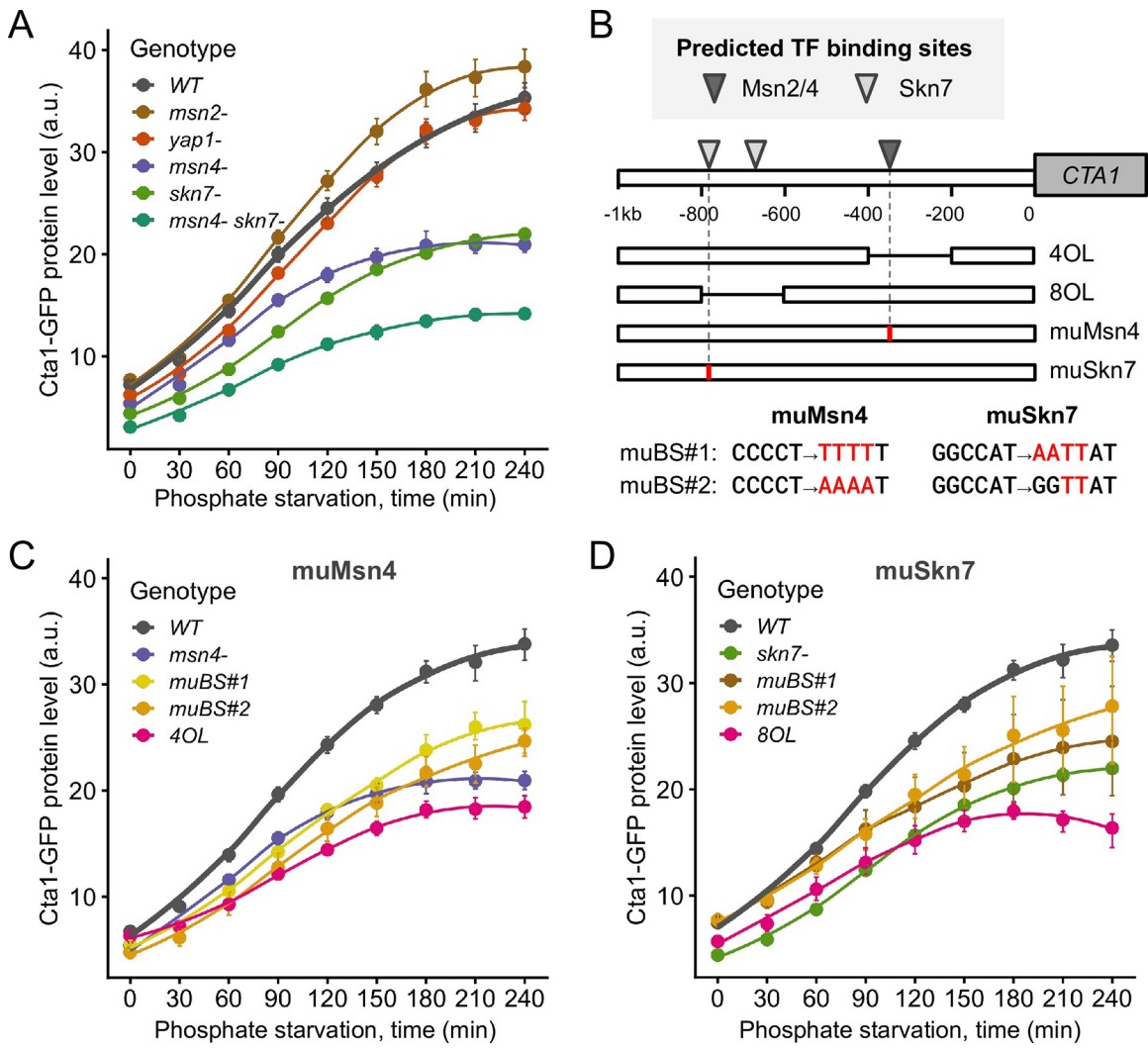

**Fig 4. TFs CgMsn4 and CgSkn7 jointly contribute to *CTA1* induction during phosphate starvation in *C. glabrata*.** (A) Cta1-GFP levels over 4 hours of phosphate starvation for wild type (WT) and TF deletion mutants. The dots represent the mean of > 3 biological replicates and the error bars indicate the 95% confidence interval based on 1000 bootstrap replicates. The lines are LOESS fit to the data. (B) A schematic showing the predicted binding sites (BS) for Skn7 and Msn2/Msn4 in *C. glabrata* (Materials and Methods). Based on the prediction, two types of *cis*-mutants were created: the internal deletions (4OL, 8OL) each removed 200 bp containing either the predicted Msn2/Msn4 BS or both Skn7 BS; four point mutation alleles each targeted either the single Msn2/Msn4 BS or one of the Skn7 BS predicted to have a higher affinity. (C, D) Comparing the Cta1-GFP induction driven by each of the mutant promoters in (B) compared with the WT and the corresponding TFΔ mutant.

Msn4 and Skn7 were required for *CTA1* induction (Fig 4A). However, since Cta1-GFP was still induced in the *msn4Δ skn7Δ* strain, additional TFs must be involved.

We also measured Cta1 induction during $H_2O_2$ treatment and confirmed that both Yap1 and Skn7 were required, while *msn4Δ* resulted in a slight decrease in induction and *msn2Δ* had no measurable effect under the $H_2O_2$ concentration tested (S7 Fig). This showed that the induction of the same effector gene, *CTA1*, depended on different TF combinations under different stresses, a result consistent with previous findings in *S. cerevisiae* [10].

To determine if Msn4 and Skn7 directly regulate *CTA1* under phosphate starvation, we first predicted their binding sites in the promoter of *CTA1* (Fig 4B). This allowed us to construct a series of promoter mutants, including two internal deletions (200 bp) removing the predicted

Msn4 or Skn7 binding sites, as well as point mutations predicted to abrogate Msn4 and Skn7 binding (Fig 4B). For both TFs, we found that all promoter mutants exhibited reduced Cta1-GFP levels similar to the TF deletion strain, with the 200 bp deletion mutants showing the most severe phenotypes (Fig 4C and 4D). These results strongly suggest that Msn4 and Skn7 directly regulate *CTA1* during phosphate starvation. In summary, we found that Msn4 and Skn7, along with unidentified TFs, jointly regulate *CTA1* induction in *C. glabrata* under phosphate starvation, while Yap1 and Skn7 were the main TFs responsible for its induction under $H_2O_2$ stress.

## Msn4 translocates into the nucleus upon phosphate starvation in *C. glabrata*

The activity of Msn4 and its paralog Msn2 are partially regulated via nuclear translocation in both *C. glabrata* and *S. cerevisiae* [40–43]. We therefore asked if Msn4 orthologs respond differently to phosphate starvation in *C. glabrata* vs *S. cerevisiae*. Using GFP-tagged Msn4 and fluorescent microscopy, we found that phosphate starvation led to 47% of the *C. glabrata* cells to have nuclear localized Msn4 (Msn4$^{nuc}$), compared with $< 4\%$ in *S. cerevisiae* (Fig 5, Bonferroni corrected $P < 0.01$). This divergence is condition-specific: glucose starvation led to a similar level of Msn4$^{nuc}$ in both species (Fig 5, 76% in *C. glabrata* vs 73% in *S. cerevisiae*, raw $P = 0.7$). No significant difference was observed between species under the no-stress condition (raw $P = 1$). These results further support CgMsn4 as one of the TFs mediating the phosphate starvation induced ASR in *C. glabrata* and that divergence in how Msn4 responds to phosphate starvation is a major contributor to the species divergence in ASR. In contrast, we found that the paralog Msn2 translocated into the nucleus in a comparable fraction of cells in the two species under phosphate starvation: 52% in *C. glabrata* and 54% in *S. cerevisiae* (S8 Fig, $P = 0.8$). Combined with the lack of effect of *msn2Δ* on Cta1 induction, we conclude that Msn2 is not involved in the ASR and doesn't contribute to its divergence.

## The Greatwall kinase, Rim15, is an important signaling component in the ASR network

Given Msn4's role in the phosphate starvation-induced ASR in *C. glabrata* (Figs 4 and 5), we used it as a bait to identify upstream signaling components in the ASR network. In *S. cerevisiae*, a key regulator of Msn2/Msn4 activity is the Greatwall kinase, Rim15 [44,45]. Rim15 itself is regulated by multiple nutrient and stress-sensing kinases, e.g., TORC1, PKA and Pho80/85, making it a hub for integrating nutrient and stress signals [46,47]. We hypothesize that Rim15 regulates the nuclear localization of Msn4 during the primary stress in *C. glabrata* (Fig 6A). To test this, we compared the nuclear localization of CgMsn4-GFP in the wild type and *rim15Δ* backgrounds. We found that ~68% of the wild type cells had nuclear-localized CgMsn4 during phosphate starvation, compared with only 22% in the *rim15Δ* background (Fig 6B, $P < 0.001$). By contrast, both strain backgrounds showed a low fraction of cells with CgMsn4$^{nuc}$ under high phosphate conditions (Fig 6B, $P = 0.2$). Next, we asked if Rim15 is required for *CTA1* induction. We found that *rim15Δ* reduced Cta1-GFP induction to a similar level as *msn4Δ*, supporting its role as a regulator of the ASR (Fig 6C). *msn4Δ rim15Δ* had a more severe effect than either deletion alone, suggesting additional players besides the Rim15-Msn4 pathway (Fig 6C). Lastly, to assess the role of Rim15 in the phosphate starvation-induced ASR, we performed quantitative CFU assays to compare the wild type and *rim15Δ* strains at a comparable strength of $H_2O_2$ (Fig 6D). While there is a significant ASR effect in both strains (both raw $P = 0.008$), the ASR effect is weaker in *rim15Δ* cells (mean ASR-score = 3.45 for *rim15Δ*, 95% CI = [2.54, 4.33]; mean ASR-score = 6.29 for the WT, 95% CI = [4.58, 8.53], Mann-Whitney U

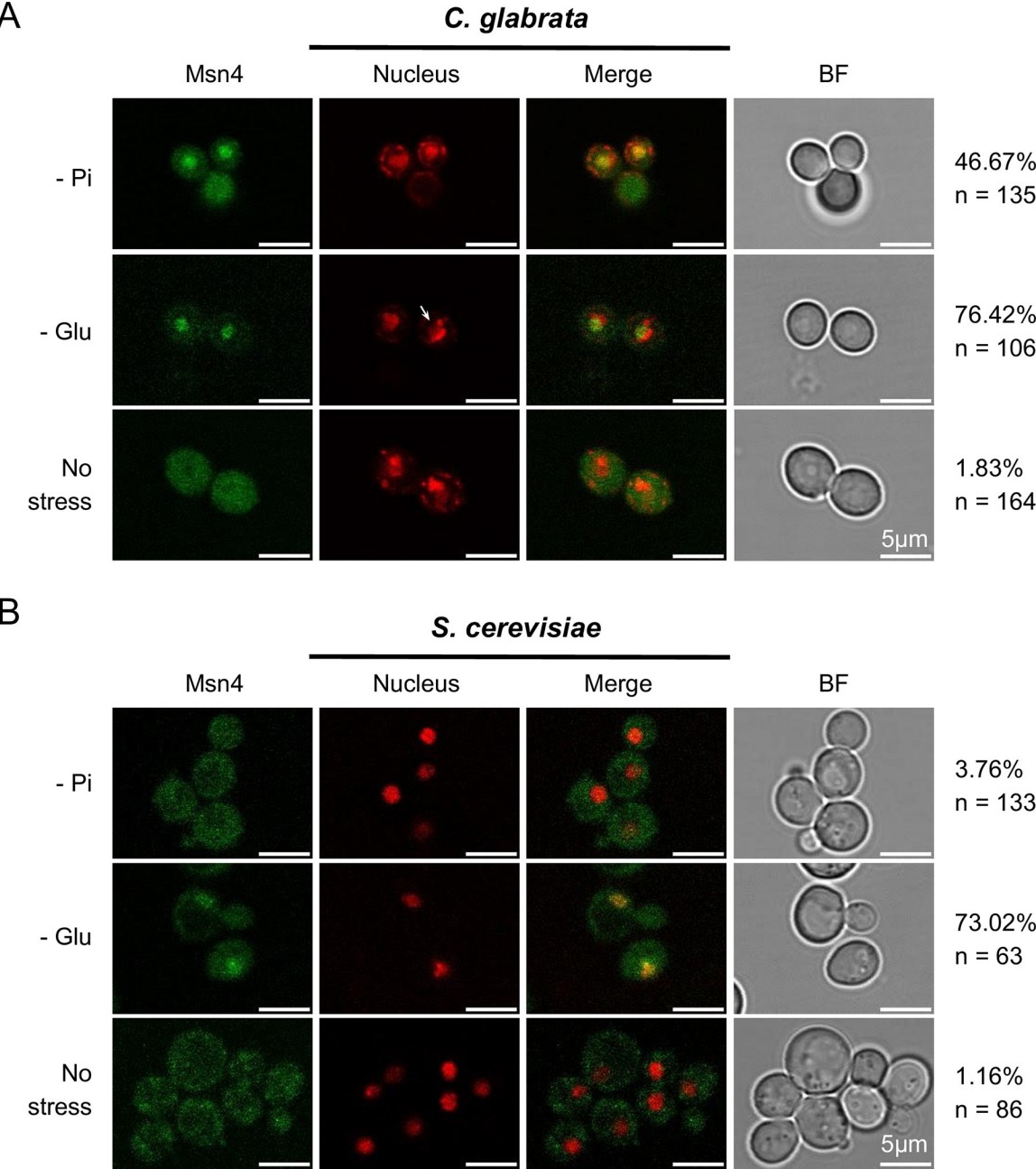

**Fig 5. *C. glabrata* Msn4 (CgMsn4) translocates into the nucleus upon phosphate starvation but not its ortholog in *S. cerevisiae* (ScMsn4).** Cellular localization of CgMsn4 (A) and ScMsn4 (B) under phosphate starvation (0 mM Pi, -Pi), glucose starvation (0.02% glucose, -Glu), and no stress conditions. All treatments were for 45 minutes. From left to right: i. CgMsn4-yeGFP and ScMsn4-mCitrine; ii. nucleus staining with DAPI in *C. glabrata* or labeled with Nh6a-iRFP in *S. cerevisiae*; iii. merge; iv. bright field. The percent of cells with nuclear-localized Msn4 and the total number of cells examined by microscopy were shown on the right. Scale bars are 5 μm in length.

test between genotypes, two-sided *P* = 0.04). This ASR defect was rescued by putting *RIM15* back to its endogenous locus (S9 Fig). Taken together, these results strongly suggest that Rim15 is a crucial signaling component for the phosphate starvation-induced ASR in *C. glabrata*.

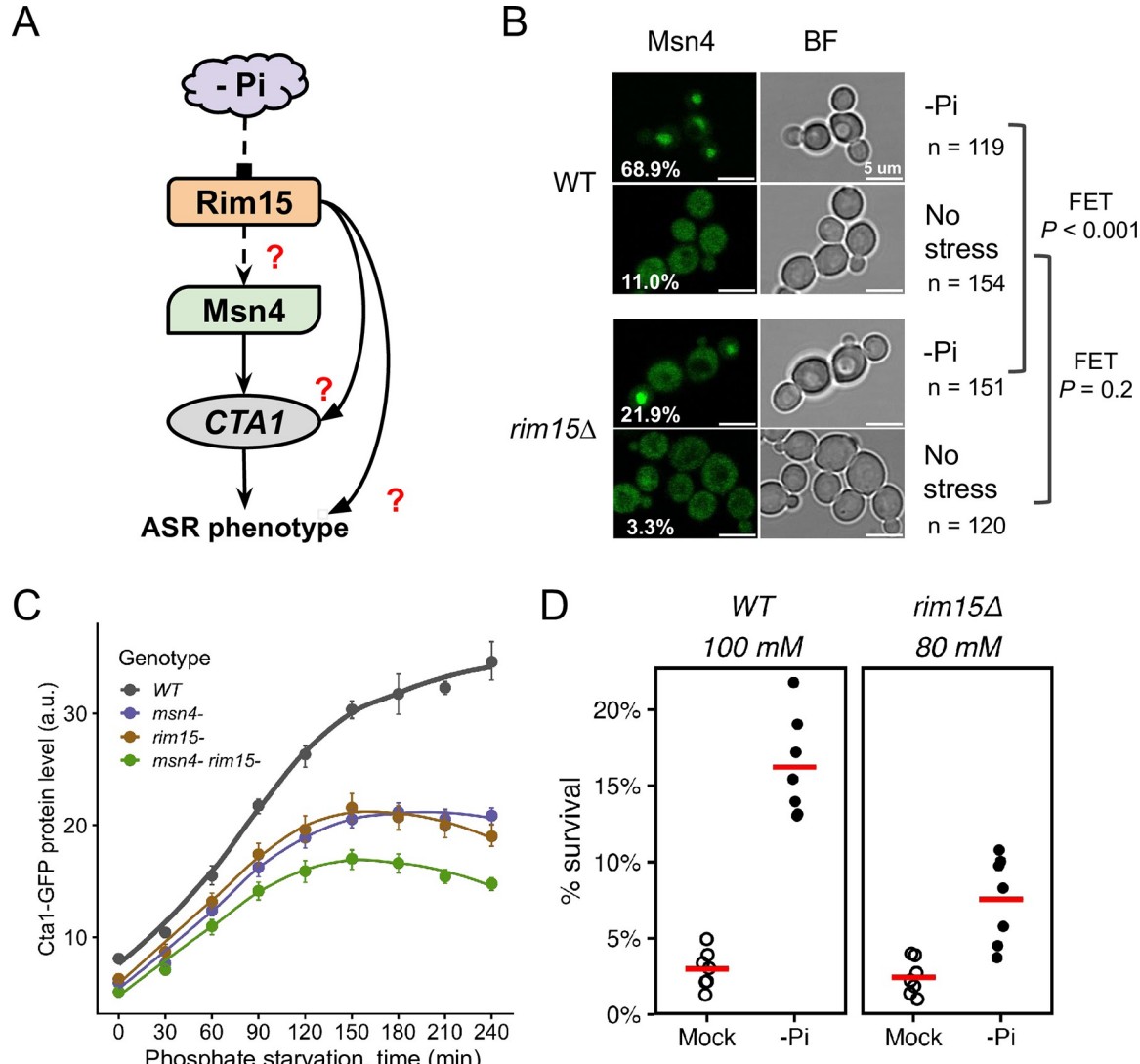

**Fig 6. The Greatwall kinase homolog Rim15 mediates phosphate starvation-induced ASR in *C. glabrata*.** (A) Proposed model for phosphate starvation-induced ASR in *C. glabrata*: phosphate starvation (-Pi) activates Rim15, which goes on to regulate Msn4's activity and contributes to ASR. Dashed arrows show proposed connections; solid arrows are regulations supported by previous results; red question marks indicate specific downstream effects to be tested. (B) CgMsn4 nuclear localization in WT and *rim15Δ* strains in -Pi or rich media. Left: CgMsn4-GFP, % of cells with CgMsn4nuc labeled on the lower left; right: bright field (BF) showing cells. Fisher's Exact Tests (FET) were performed to compare the two strains under each condition. Raw, two-sided *P*-values were reported on the side. (C) Cta1-GFP induction during -Pi. Dots are the means of >3 biological replicates and the error bars the 95% CI based on 1000 bootstraps. (D) ASR assay for the wild type (WT) and *rim15Δ*. Shown are survival rates of the two strains at an equivalent $H_2O_2$ strength, either with (-Pi) or without (Mock) a primary stress. Each dot is an independent biological replicate (n = 7) and the red bar represents the mean. Basal survival rates were not different between the two strains (*P* = 0.5). The ASR effect is significant in both (raw, one-sided *P* = 0.008 for both), but is significantly lower in *rim15Δ* (mean ASR-score = 6.29 in WT vs 3.45 in *rim15Δ*, Mann-Whitney U test, two-sided *P* = 0.04).

## Target-Of-Rapamycin Complex 1 (TORC1) is strongly inhibited by phosphate starvation in *C. glabrata* but not in *S. cerevisiae*

In *S. cerevisiae*, Msn2 and Msn4's nuclear translocation is regulated by both the Target-of-Rapamycin Complex 1 (TORC1) and Protein Kinase A (PKA) pathways [41,45,48]. In *S. cerevisiae*, TORC1 responds slowly and weakly to phosphate limitation [49]. In the distantly

related yeast pathogen *C. albicans*, TORC1 was shown to respond rapidly to phosphate availability [28]. We therefore hypothesized that TORC1 is more strongly and quickly inhibited by phosphate starvation in *C. glabrata* than in *S. cerevisiae*, contributing to the Msn4-mediated stress response and ASR. To test this, we used the phosphorylation of ribosomal protein S6 (Rps6) as a conserved readout of TORC1 activity [28,50,51]. We found that 60 minutes of phosphate starvation did not affect the total level of Rps6 but markedly reduced the abundance of phosphorylated Rps6 (P-Rps6) in *C. glabrata* (Bonferroni-corrected $P = 0.06$, Fig 7A and 7B). The same treatment, however, did not affect the proportion of P-Rps6 in *S. cerevisiae* ($P = 1$). By contrast, nitrogen starvation rapidly inhibited phosphorylation of Rps6 in both species ($P = 0.06$ and $0.07$). These results confirmed that inhibition of TORC1 by nitrogen starvation is conserved between the two species while only in *C. glabrata* is TORC1 strongly inhibited by phosphate starvation at 60 minutes, consistent with their distinct ASR phenotypes (Fig 1). To determine which of the two species represents the ancestral state, we performed the same P-Rps6 assay in two outgroup species, *K. lactis* and *L. walti*, which diverged from *S. cerevisiae* and *C. glabrata* approximately 120 million years ago [15]. Both species exhibited intermediate levels of TORC1 inhibition by phosphate starvation at 60 minutes, suggesting that phosphate signaling to TORC1 is ancestral, but the signaling strength has evolved (S10 Fig). By contrast, in all four species TORC1 was rapidly inhibited by nitrogen starvation, suggesting that nitrogen is an evolutionarily conserved input for TORC1.

## Inhibition of TORC1 leads to Msn4 nuclear localization, *CTA1* induction and confers mild ASR in *C. glabrata*

To determine if TORC1 inhibition can lead to ASR, we used rapamycin to specifically inhibit TORC1 and monitored Msn4[nuc], Cta1-GFP induction and the ASR effect. We found that TORC1-inhibition induces Msn4[nuc] in both species, with a stronger effect in *C. glabrata*: 45 min of exposure to 500 ng/mL of rapamycin causes 71.8% (79/110) of *C. glabrata* cells and 14.5% (19/131) of *S. cerevisiae* cells to have Msn4[nuc]. Both were significantly higher compared with the mock-treated group (1.8% in *C. glabrata* and 1.2% in *S. cerevisiae*, raw FET $P < 0.001$ in both). Next, we followed Cta1-GFP induction in *C. glabrata* exposed to rapamycin and found a dose-dependent response except for the highest dose, which resulted in a lower induction than the next dose (Fig 7C). Lastly, we found 125 ng/mL rapamycin treatment for 45 minutes modestly but significantly enhanced the survival of *C. glabrata* following a secondary $H_2O_2$ stress (Fig 7D, mean ASR-score = 1.79, 95% CI [1.35, 2.28], raw $P = 0.016$). The same rapamycin dose resulted in a similar increase in survival in *S. cerevisiae* (mean ASR-score = 1.81, 95% CI [1.21, 2.47], raw $P = 0.063$) (Fig 7D). We conclude that TORC1 inhibition has the potential to induce acquired resistance for $H_2O_2$ in both species. This implies that divergence of the ASR occurred through TORC1's differential response to -Pi.

## TORC1 proximal effector, Sch9, contributes to phosphate starvation-induced ASR, while a downstream output of the Tap42-PP2A branch is minimally affected

The AGC kinase Sch9 is directly phosphorylated and regulated by TORC1 [49]. In turn, Sch9 regulates multiple downstream processes, including repressing stress responses by inhibiting Rim15 via phosphorylation, which prevents it from activating Msn4 [52,53]. We hypothesized that Sch9 negatively regulates the ASR in *C. glabrata*. To test this, we created a TORC1 phosphomimetic mutant of CgSch9, predicted to be constitutively active with respect to TORC1 signaling [49] (S11 Fig). This mutant showed reduced Cta1-GFP induction during phosphate starvation (Fig 7E). Moreover, a quantitative CFU assay failed to detect ASR in this mutant but

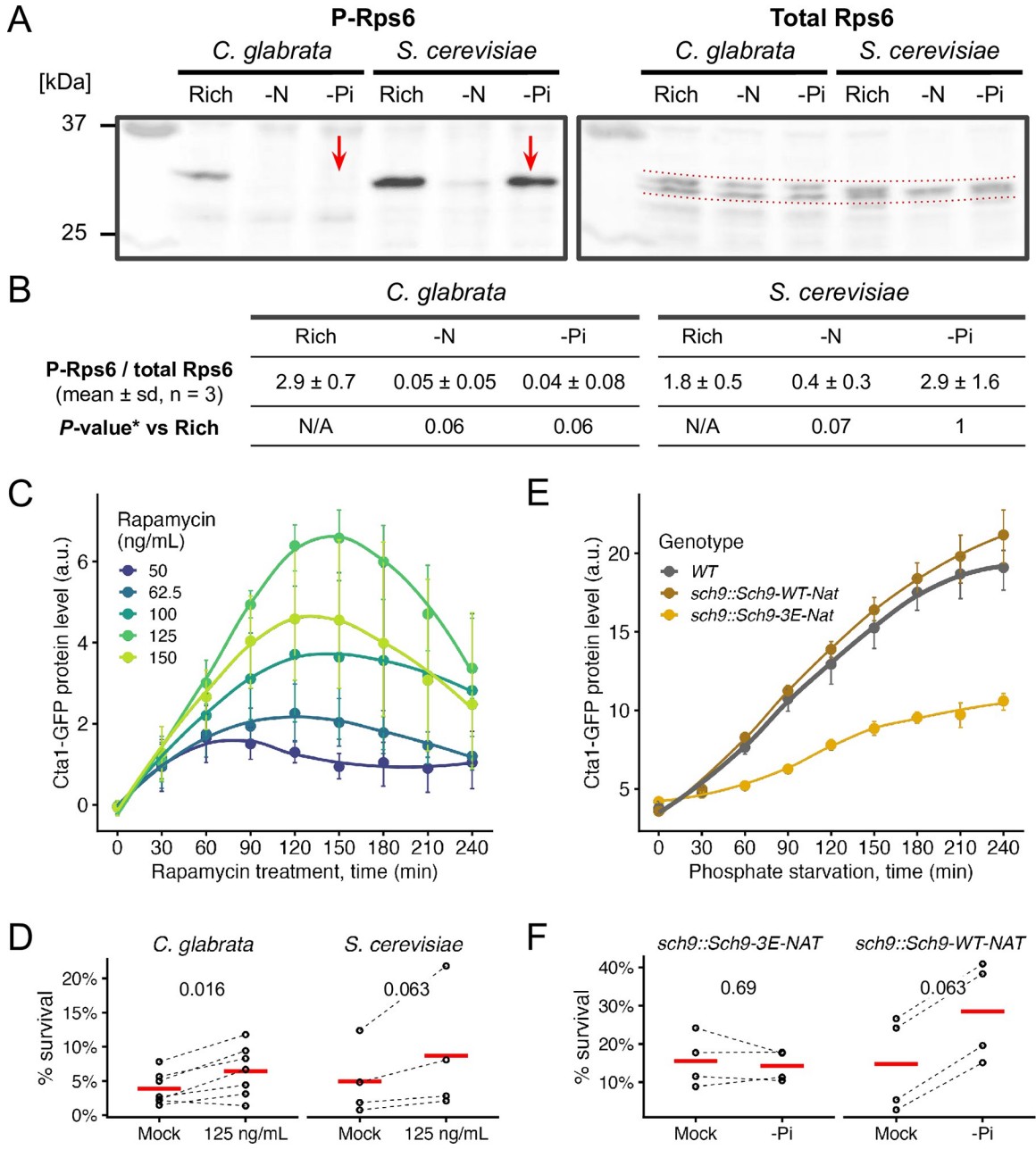

**Fig 7. TORC1 is strongly inhibited by phosphate starvation in *C. glabrata*, likely contributing to the ASR via its proximal kinase, Sch9.** (A) A representative Western Blot for phosphorylated Rps6 (P-Rps6) and total Rps6 in both species under rich media, nitrogen starvation (-N) or phosphate starvation (-Pi) conditions. Red arrows point to the loss vs presence of the P-Rps6 band under -Pi in *C. glabrata* and *S. cerevisiae*, respectively. (B) Quantification of %P-Rps6 over total Rps6 (n = 3). *Bonferroni corrected *P*-values from Student's t-tests were shown. (C) Inhibiting TORC1 by rapamycin induces Cta1-GFP in a dose-dependent manner. The dots, error bars and lines have the same meaning as before. (D) ASR for $H_2O_2$ with rapamycin as the primary treatment. Plotted are survival rates with or without rapamycin treatment (n = 12 for *C. glabrata*, n = 8 for *S. cerevisiae*). 60 mM and 6 mM of $H_2O_2$ were used as the secondary stress for the two species, resulting in a similar basal survival rate (*P* = 0.6). A Wilcoxon signed-rank test was used to compare the paired experiments with or without the primary treatment for each species. The raw, one-sided *P*-values were shown on the top. (E) Same plot as C, comparing Cta1-GFP induction in a phosphomimetic mutant of Sch9, a key proximal effector of TORC1, and a matching wild type Sch9 strain. (F) Same as D but with phosphate starvation (-Pi) as the primary stress, comparing the Sch9-3E mutant (n = 4) and the matching Sch9-wt control (n = 4). 100 mM and 40 mM of $H_2O_2$ were used as the secondary stress for the two genotypes, resulting in a similar basal survival rate (*P* = 0.7).

did in the matching complement strain (Fig 7F, raw $P$ = 0.69 and 0.063, respectively), further supporting TORC1's role in the ASR.

Since TORC1 coordinates multiple cellular processes, we asked if phosphate starvation had limited or broad effects on TORC1's downstream branches [54]. To test this, we monitored the Nitrogen Catabolism Repression (NCR) regulon, which is controlled by a different branch, Tap42-PP2A, rather than by Sch9, and is differentially induced under nitrogen starvation vs glucose starvation and other stresses in *S. cerevisiae* [55]. Using an established *DAL80pr*-GFP reporter [56], we found that nitrogen starvation strongly induced NCR-sensitive genes, but phosphate starvation had minimal effects in both *S. cerevisiae* and *C. glabrata* (S12 Fig). This suggests that evolution can wire an input such as phosphate limitation to a specific TORC1 output, e.g., stress response, without strongly impacting other outputs.

## Discussion

Acquired stress resistance (ASR) is suggested to be an adaptive anticipatory response [7]. As such, it is expected to evolve between species. However, both the genetic basis of ASR and its evolution remain poorly characterized. In this study, we discovered a divergent ASR phenotype between two related yeasts: in the opportunistic pathogen, *C. glabrata*, a short, non-lethal phosphate starvation provides a strong protective effect for a severe $H_2O_2$ stress, while in the related baker's yeast, *S. cerevisiae*, the same treatment has little to no effect (Fig 1). We identified a core subnetwork behind the ASR in *C. glabrata*, where phosphate starvation not only induces the canonical phosphate starvation (PHO) response via the TF Pho4, but also induces more than 15 oxidative stress response (OSR) genes, mediated by two TFs involved in the canonical OSR, Msn4 and Skn7 (Figs 2 and 8A). Divergence in ASR between the two species is due in part to the difference in nuclear translocation of Msn4 in response to phosphate limitation. Remarkably, the central regulator TORC1 showed differential response to phosphate in the two species while being similarly affected by nitrogen starvation (Fig 7). This implicates TORC1 as a key point of divergence behind the ASR evolution; it also shows how a conserved, multifunctional complex can be evolutionarily tuned.

We showed that the divergence in the ASR for $H_2O_2$ between the two species is specific to phosphate–two other primary stresses tested, glucose starvation and heat shock, elicit similar protective effects in both species (S2 Fig). Why phosphate? Several lines of evidence suggest that phosphate and genes involved in the PHO response are relevant in the host environment, including impacting virulence in several infection models and affecting sensitivity to ROS [27,57]. Moreover, while studies on the phagosome environment mainly focused on nutrients such as glucose, a recent transcriptomic study showed that PHO genes, such as the high affinity transporter *PHO84* and the secreted phosphatase *PMU2* were both highly upregulated within 2 hours of *C. glabrata* being engulfed by human macrophages (S13 Fig) [31]. Conversely, we found that many genes induced when *C. glabrata* cells were engulfed by macrophage were also induced during phosphate starvation, including genes involved in autophagy, TCA cycle, amino acid biosynthesis and iron homeostasis (S14 Fig). These results suggest that phosphate limitation is physiologically relevant in the host and may serve as a stimulus to induce ASR for more severe stresses.

Does phosphate starvation provide general protection against any stress or is it specific to certain types of stresses? We found that the protective effect of phosphate starvation varies depending on the types of ROS used as the secondary stress (S3 Fig): the protective effect is the strongest for $H_2O_2$, moderate for menadione sodium bisulfite, and non-existent for tBOOH. The lack of ASR for tBOOH, an alkyl hydroperoxide, was initially surprising. However, this can be understood based on its many differences from $H_2O_2$, e.g., its main cellular targets are

lipids in the plasma membrane; it reacts differently with the cellular redox system, and elicits different signaling in *S. cerevisiae* than $H_2O_2$ [58–60]. This demonstrates that ASR is specific to the primary and secondary stress combinations. We further hypothesize that these combinations are frequently encountered in the organism's environment, providing the selective pressure for the ASR.

The ASR effect in *C. glabrata* bears similarity to a previously described stress resistance phenotype following chronic starvation in *S. cerevisiae* [61–65]. Could the divergence in ASR be attributed to the difference in timing/threshold of the chronic starvation response? We think not. First, the chronic starvation response is associated with severe cell cycle arrests. We found that during a short (< 2hrs) phosphate starvation, both species continue to divide with a similar fraction of unbudded cells, suggesting that cell-cycle arrest cannot explain the species difference in ASR (S15 Fig). Moreover, while several OSR genes were induced during chronic phosphate starvation in *S. cerevisiae*, catalase genes were notably not among them [62]. This is in contrast to the crucial role of *CTA1* induction for the ASR in *C. glabrata*. We therefore believe that the phosphate starvation-induced ASR has distinct transcriptional basis and serves a different purpose than the chronic starvation response.

Does ASR involve the same effector and regulator genes as the canonical response to the secondary stress, or is it made up of an entirely different set of genes? We found phosphate starvation-induced ASR in *C. glabrata* shares many of the TFs and effector genes with the canonical OSR, but differs in the TF combinations required to induce the effector genes such as *CTA1*, resulting in distinct induction kinetics (Figs 2, 3 and 8B). Such condition-specific regulation of stress response genes has also been observed in *S. cerevisiae* and may be a hallmark of stress response networks in general [10]. In theory, this could facilitate the evolution of stress responses, including the ASR, by allowing for the reuse of existing genes and networks. This likely depends on cis-regulatory mutations that modify the expression pattern of a gene in one context with little or no impact under another, thereby reducing the pleiotropic effects. Another intriguing finding is that the paralogous Msn2 and Msn4 showed divergent functions in ASR and $H_2O_2$ stress between species. In *S. cerevisiae*, the two paralogs have largely identical functions, with Msn2 playing a more important role [32,66–68]. By contrast, Msn2 is dispensable for the induction of *CTA1* in *C. glabrata* both during phosphate starvation and $H_2O_2$ stress, while Msn4 plays an important role (Figs 4, 5 and S7). We speculate that the maintenance and evolution of TF paralogs is an important source for ASR and, more generally, stress response evolution.

Lastly, given the central role and conservation of TORC1 across species, it is surprising that its response to phosphate differed significantly between the two yeasts, which likely led to their ASR divergence. How can a central, multi-functional regulator like TORC1 evolve? Our results show that phosphate starvation specifically induced the stress response branch of TORC1 while it had minimal effects on another branch associated with nitrogen catabolism (Figs 7 and S12). We propose that the ability to rewire specific input and output branches of TORC1 allows evolution to fine-tune the regulator's function without having pleiotropic effects (Fig 8C).

## Materials and methods

### Experimental reproducibility and data availability

Each experiment was repeated at least two but mostly three or more times with >2 biological replicates each. Data analysis and visualization were performed in R v4.2. Raw data, R markdown files, and output files are available at https://doi.org/10.5281/zenodo.10004973. RNA-seq data for *C. glabrata* before and after 1 hour of phosphate starvation is available through the GEO database (https://www.ncbi.nlm.nih.gov/geo/query/acc.cgi?acc=GSE244380).

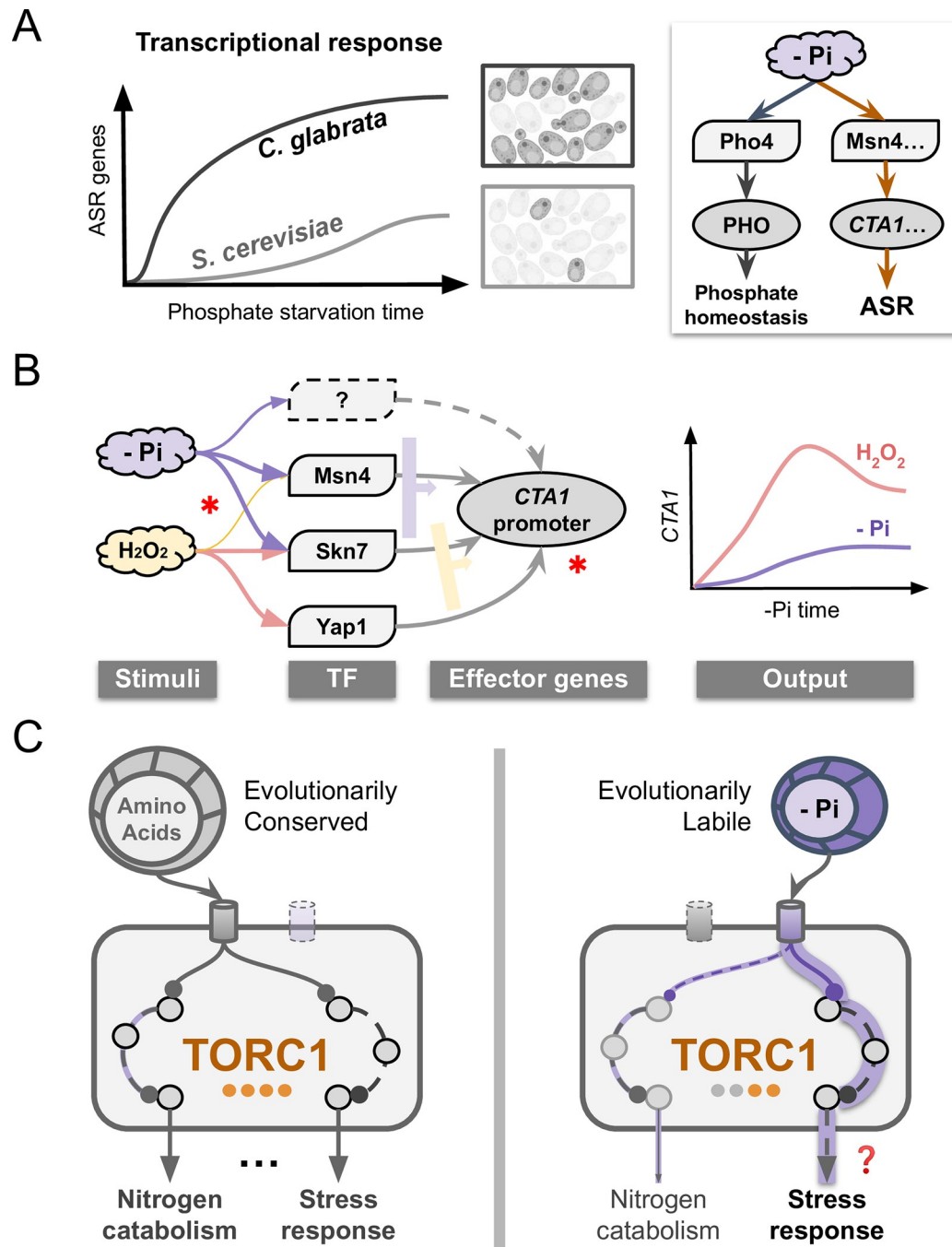

**Fig 8. ASR divergence and rewiring of the underlying regulatory network.** (A) Phosphate starvation elicits a fast and strong induction of oxidative stress response (OSR) genes in *C. glabrata* in addition to the canonical phosphate starvation response (right), providing strong acquired resistance for a secondary $H_2O_2$ challenge; in the related *S. cerevisiae*, the induction of OSR genes is much weaker and slower, explaining its lack of ASR. The orange arrows in the right diagram indicate evolutionary rewiring in part of the response to phosphate limitation between the two species. (B) Evolutionary rewiring at the transcriptional level. Regulation of OSR genes such as *CTA1* involves some of the same transcription factors (TFs) during both oxidative stress and phosphate starvation, but with different combinatorial logic. Width of the arrow indicates the importance of the TF; the wide arrows to the right of the TFs indicate their combinatorial logic. Asterisks indicate potential evolutionary events, both at the *trans*- and *cis*- (promoter) levels. Combined, they lead to distinct induction kinetics under the two stimuli (right). (C) Proposed rewiring in the Target-of-Rapamycin Complex 1 (TORC1). Nitrogen-sensing is conserved between species and strongly activate two of TORC1's downstream branches. By contrast, sensing of phosphate is evolutionarily labile; it strongly activates the stress response branch but very weakly affects the nitrogen catabolism branch. The question mark indicates we still lack direct evidence for TORC1 being responsible for the

stress gene induction in the ASR. This model suggests that flexibility in connecting individual stimulus with specific downstream branch(es) allows TORC1 to contribute to the ASR evolution by avoiding pleiotropic effects.

## Yeast media and growth conditions

Yeast cells were grown in the Yeast extract-Peptone Dextrose (YPD) medium or the Synthetic Complete (SC), using Yeast Nitrogen Base without amino acids (Sigma Y0626) supplemented with 2% glucose and amino acid mix. Unless specified, all stress treatments were performed with mid-log phase cells. Briefly, cells were grown overnight in either YPD or SC media, diluted in the morning to $OD_{600}$ = 0.2 and grown to $OD_{600}$ ~ 1. To apply the treatments, cells were collected by centrifugation at 3,000g for 5 minutes, washed 2–3 times with water and released into the treatment media. Nitrogen starvation medium was made using the Yeast Nitrogen Base without amino acids and without ammonium sulfate (Sigma, Y1251). Glucose starvation medium had 0.02% or 0.05% glucose in the SC medium. Phosphate starvation medium was made using Yeast Nitrogen Base with ammonium sulfate, without phosphates, without sodium chloride (MP Biomedicals, 114027812) and supplemented to a final concentration of 2% glucose, 1.5 mg/ml potassium chloride, 0.1 mg/ml sodium chloride and amino acids, as described previously [25]. Phosphate concentration in the medium was measured using a Malachite Green Phosphate Assay kit (Sigma, MAK307). The -MC medium used to induce *MET3pr-CTA1* was made the same way as the SC medium, except methionine and cysteine were omitted from the amino acid mix. To test the effect of inhibiting TORC1 on ASR, Rapamycin (RPI, R64500) was added into the SC medium at the indicated concentrations. To apply $H_2O_2$ stress, $H_2O_2$ stock solution (30% w/w, Sigma H1009) was diluted into SC medium to the indicated concentrations immediately before the treatment, except in the *MET3pr-CTA1* experiment, where -MC medium was used instead of SC medium to allow *CTA1* to be induced during the secondary stress. To test additional ROS for the ASR effect, tert-butyl hydroperoxide (tBOOH 70% w/w in $H_2O$, Sigma 458139) was diluted directly into the SC medium, while menadione sodium bisulfite (Sigma M5750) was first dissolved in $ddH_2O$ to make a 1M stock, and then the stock was diluted into the SC medium at the indicated concentrations immediately before the treatment.

## Generating yeast strains and plasmids

Genetic transformation of both *S. cerevisiae* and *C. glabrata* was performed using the LiAc method as described in [69]. *C. glabrata* knockout strains were generated with auxotrophic markers *HIS3* and *URA3* and the nourseothricin drug marker NAT. ~200 bp of flanking homology sequences were added to either a deletion cassette or a knock-in construct using overlap PCR. Transformants were confirmed by colony PCR on both sides of the gene. To construct *CTA1* promoter mutants, a URA3 cassette was used to replace the endogenous promoter, after which the mutant allele constructed by PCR mutagenesis was used as the repair template to swap out the URA3 and selecting with 5-fluoroorotic acid (5-FOA) (GoldBio, F-320). A similar approach was used to replace the endogenous *CTA1* promoter with the promoter from the *C. glabrata MET3* gene. The *CgMET3* promoter was amplified from pCU-MET3 [70] (AddGene #45336). The Msn2 and Msn4-GFP expression plasmids were made by inserting the coding sequence of the genes in between the *PDC1* promoter and GFP in the pCU-PDC1-GFP vector [71] (AddGene #45342). The *ScDAL80*-GFP reporter was constructed by amplifying the 1kb upstream region of *ScDAL80* and placing it in front of a yeGFP in a pRS426 (ATCC #77107) backbone. Gibson cloning was performed according to the protocol of the Gibson Assembly kit (NEB, E5520). To construct the Sch9-3E-NAT mutant and the

matching Sch9-WT-NAT control strains, we first used PCR-based mutagenesis to construct the Sch9-3E fragment (S11 Fig). This and the Sch9-WT fragments were joined with the NAT cassette via overlap PCR. Then, ~200bp of flanking sequences homologous to the endogenous *SCH9* 5' and 3'UTR were added on both sides using overlap PCR. The end product was used as a linear template to transform an *SCH9* wild type strain of *C. glabrata*. Positive transformants were selected on YPD+NAT plates and validated by genomic PCR and sequencing. Yeast strains, plasmids and qRT-PCR primers used in this study are listed in Tables 1, 2 and 3, respectively.

### Spotting and Colony Forming Units (CFU) assays

A semi-quantitative spotting assay or a quantitative Colony Forming Unit (CFU) assay was used to measure the survival rate after a treatment. For the spotting assay, the treated cells were 10x serially diluted in ddH₂O. 5 μl of each dilution was spotted onto an SC plate and allowed to dry before the plate was incubated at 30C for 16–36 hours. An image was taken at the end of the incubation using a homemade imaging hood fitted with an iPad Air. For the CFU assay, post-treatment cells were diluted 100 to 1,000-fold and plated on an SC plate. The dilution was chosen to produce roughly 10–500 colonies per plate. The plates were incubated for 48 hours and CFU was manually counted using a lab counter (Fisher Scientific).

### Calibrating $H_2O_2$ concentrations

To apply an equivalent strength of $H_2O_2$ treatment to species and genotypes with different basal survival rates, we measured the cell survival rates after treatment with different concentrations of $H_2O_2$. Specifically, we exposed *S. cerevisiae* cells to a series of $H_2O_2$ concentrations from 0mM to 10mM, with 2mM steps. Similarly, we exposed *C. glabrata* cells $H_2O_2$ concentrations from 0mM to 100mM with 20mM steps. CFU assays were performed after 2 hours of treatment to estimate the survival rate. A Mann-Whitney U test was used to determine the significance of differences. Concentrations resulting in similar survival rates across species or genotypes were chosen as the condition for the secondary stress in the ASR experiment.

### Acquired stress resistance (ASR) assay

During the primary stress phase, mid-log phase cells were collected, washed, and released into either the SC medium for a mock treatment or the primary stress media, e.g.,—Pi for phosphate starvation. After 45 minutes (or as indicated) of incubation with shaking at 30˚C, cells were collected by centrifugation and resuspended at $OD_{600}$ = 0.2 in the secondary stress treatment media, then incubated the same as before for 2 hours. After the secondary stress, cells were diluted and directly used for the spotting or CFU assay. CFU multiplied by the dilution rate was then used to calculate r (MO/MM), r' (PO/PM) and the ASR-score (r'/r). Three to eight biological replicates were performed.

### RNA-seq profiling for *C. glabrata* under phosphate starvation

*C. glabrata* wild type cells were grown in SC overnight, diluted into fresh SC the next morning to $OD_{600}$ = 0.1, grown for another 2–3 hours until the culture reached mid-log phase ($OD_{600}$~0.6). At this point, two biological replicates of pre-starvation samples were collected using a cold methanol quenching method [35,72]. Briefly, 5 mL of culture was added directly into 7.5 mL of pre-chilled methanol (−50˚C) and incubated in an ethanol-dry ice bath at that temperature for at least 20 min. The remaining culture was collected by filtration, washed with equal volume of pre-warmed no phosphate SC media and then released into the pre-warmed

**Table 1. Yeast strains used in this study.**

| Strain name | ID | Genotype | Source | Figure |
|---|---|---|---|---|
| *C. glabrata* BG99 | yH181 | [BG2] his3Δ(1 + 631) | [79] | Figs 7, S2, S3, S4, S10 |
| *C. glabrata* BG99 derivative | yH001; yH002 | [BG2] ura3::pUC19; his3Δ(1+631) | [25] | Figs 1, 3, 5, 6, 7, S1, S4, S6, S8, S9, S10, S15 |
| *S. cerevisiae* K699 | yH154 | [W303] *MATa ade2-1 trp1-1 can 1–100 leu2-3,112 his3-11,15 ura3* GAL+ | ATCC #200903 | Figs 1, 6, 7, S1, S2, S10 |
| *S. cerevisiae* S288C | yH545 | [S288C] *MATα his3Δ200 leu2Δ1 ura3-52 trpΔ63 lys2Δ202 canR cyhR* | Gift from Dr. Jan Fassler | S12, S15 Figs |
| *K. lactis* | yH149 | [NRRL Y-1440] wild type | ATCC #8585 | Fig S10 |
| *L. waltii* | yH217 | [NCYC 2644] wild type | Gift from Dr. Chris Hittinger | S10 Fig |
| *CTA1-GFP* | yH298; yH299 | [BG99] *cta1::CTA1-GFP* | This study | Figs 3, 4, 6, 7, S4, S5, S7 |
| *MET3pr-CTA1-GFP* | yH737 | [BG99] *MET3pr::CTA1-GFP* | This study | S5 Fig |
| *cta1Δ* | yH271; yH272 | [BG99] *cta1::URA3* | This study | Figs 3, 4, S3, S4, S6 |
| *cta1::CTA1* | yH285; yH286 | [BG99] *cta1::CgCTA1* | This study | S3, S4, S6 Figs |
| *msn2Δ* | yH417; yH418; yH419 | [BG99] *msn2::HIS3; cta1::CTA1-GFP* | This study | Figs 4, S7 |
| *msn4Δ* | yH429; yH430; yH431 | [BG99] *msn4::NAT; cta1::CTA1-GFP* | This study | Figs 4, 6, 7, S7 |
| *msn2Δ msn4Δ* | yH433; yH434; yH435 | [BG99] *msn2::HIS3; msn4::NAT; cta1::CTA1-GFP* | This study | Figs 4, S7 |
| *pho4Δ* | yH005 | [BG99] *pho4::URA3;* | This study | S6 Fig |
| *rim15Δ* | yH609; yH610 | [BG99] *rim15::HIS3;* | This study | Figs 6, S9 |
| *rim15Δ CTA1-GFP* | yH520; yH521; yH522 | [BG99] *rim15::HIS3; cta1::CTA1-GFP* | This study | Fig 6 |
| *rim15::RIM15* | yH731; yH732 | [BG99] *rim15::RIM15* | This study | S9 Fig |
| *msn4Δ rim15Δ* | yH523; yH524; yH525 | [BG99] *msn4::NAT; rim15::HIS3; cta1::CTA1-GFP* | This study | Fig 6 |
| *skn7Δ* | yH440; yH441; yH442 | [BG99] *skn7::HIS3; cta1::CTA1-GFP* | This study | Figs 4, S7 |
| *yap1Δ* | yH426 | [BG99] *yap1::Nat* | This study | S6 Fig |
| *yap1Δ CTA1-GFP* | yH446; yH447; yH449 | [BG99] *yap1::HIS3; cta1::CTA1-GFP* | This study | Figs. 4, S7 |
| *msn4Δ skn7Δ* | yH454; yH455; yH457 | [BG99] *msn4::NAT; skn7::HIS3; cta1::CTA1-GFP* | This study | S7 Fig |
| *yap1Δ skn7Δ* | yH452; yH453 | [BG99] *skn7::NAT; yap1::HIS3; cta1::CTA1-GFP* | This study | S4 Fig |
| *4OL* | yH352; yH335; yH336 | [BG99] *200bp - 400bp upstream of translational start site of CTA1 removal; cta1::CTA1-GFP* | This study | Fig 4 |
| *8OL* | yH340; yH341; yH342 | [BG99] *600bp - 800bp upstream of translational start site of CTA1 removal; cta1::CTA1-GFP* | This study | Fig 4 |
| *muMsn4BS#1* | yH353; yH354; yH355 | [BG99] *Msn4 binding site mutation 1; cta1::CTA1-GFP* | This study | Fig 4 |
| *muMsn4BS#2* | yH384; yH385; yH386 | [BG99] *Msn4 binding site mutation 2; cta1::CTA1-GFP* | This study | Fig 4 |
| *muSkn7BS#1* | yH403; yH404 | [BG99] *Skn7 binding site mutation 1; cta1::CTA1-GFP* | This study | Fig 4 |
| *muSkn7BS#2* | yH405; yH406; yH407 | [BG99] *Skn7 binding site mutation 1; cta1::CTA1-GFP* | This study | Fig 4 |
| ScMsn4-mCitrine ScMsn2-mcherry | yH223 | [W303] Nhp6a-iRFP-Kan, Msn4-mCitrine(v163A)-spHIS5 Msn2-mCherry-Trp, *leu::PRS305* | [66] | Figs 5, S8 |
| ScDAL80pr-GFP | yH646; yH647; yH648 | [BG99] *msn2::ScDal80pr-GFP* | This study | S12 Fig |
| *C. glabrata* budding assay | yH654 | [BG99] pCU (pgrb2.1) *CgPHO84p::yeGFP* | This study | S15 Fig |
| *S. cerevisiae* budding assay | yH677 | [S288C] *ura3::PHO84pr-GFP-URA3* | This study | S15 Fig |

*(Continued)*

**Table 1.** (Continued)

| Strain name | ID | Genotype | Source | Figure |
|---|---|---|---|---|
| *Sch9-WT-NAT* | yH692; yH693; yH694; yH695 | [BG99] *sch9::Sch9-WT-NAT; cta1::CTA1-GFP* | This study | Fig 7 |
| *Sch9-3E-NAT* | yH699 | [BG99] *sch9::Sch9-3E-NAT; cta1::CTA1-GFP* | This study | Fig 7 |

no phosphate SC. After 1 hour incubation with shaking at 30˚C, three biological replicates of the starved samples were collected in the same way as above. When all samples were in the ethanol-dry ice bath for > 20 minutes, cells were collected by centrifugation and quickly washed with ice-cold water to remove the methanol and resuspended in RNAlater solution (Qiagen, 76104) for at least 2 hours. Cells were centrifuged to remove the RNAlater, flash-frozen in liquid nitrogen and stored at -80 C until RNA-extraction. For each sample, ~5x10$^7$ cells were collected and total RNA was extracted using a MasterPure Yeast RNA purification kit (Biosearch Technologies, MPY03100) following the manufacturer's protocol. RNA-seq libraries were prepared with the TruSeq RNA Library Preparation Kit v2 (Illumina) with the mRNA purification option. The resulting libraries were sequenced on an Illumina HiSeq 4000, which produced on average 10 million 50 bp single end reads for each sample. Raw and processed data are available at GEO (GSE244380).

## Comparing the transcriptional response to phosphate starvation between *C. glabrata* and *S. cerevisiae*

The RNA-seq raw reads were mapped to the *C. glabrata* genome downloaded from the Candida Genome Database (CGD, RRID:SCR_002036, version s02-m02-r09), using Bowtie v1.1.1 (RRID:SCR_005476) with the option '-m 1—best–strata' [73]. The resulting SAM files were sorted using Samtools v1.2 (RRID:SCR_002105) [74] and the number of reads per transcript was counted using Bedtools2 (RRID:SCR_006646) [75], with the option 'bedtools coverage -a BAM_file -b genome_annotation.bed -S -s sorted -g Chrom.length'. Gene features for *C. glabrata* were downloaded from CGD, version s02-m07-r04. The count matrix was filtered to remove lowly expressed genes (335 genes with <1 read per sample on average). Next, we used the trimmed mean of M-values ('TMM') method in the EdgeR (RRID:SCR_012802) package to calculate the normalization factors for scaling the raw library sizes [76], and applied voom transformation [77] to remove the mean-variance relationship on the log2 transformed count data. A log2 fold change was calculated for each gene by subtracting the mean of the pre-starvation samples from each of the 1-hr starvation sample values. For *S. cerevisiae*, the microarray data from [35] was downloaded from the GEO database (RRID:SCR_005012). Within the series GSE23580, three samples corresponding to the three biological replicates comparing the 1 hour phosphate starved sample to a mock-treated sample were extracted (GSM578408, 578424, 578440). The log2 ratios were already background-subtracted, normalized, and used directly for comparisons. To compare orthologous gene expression, we first curated a list of

**Table 2. Yeast expression plasmids used in this study.**

| Plasmids | ID | Description | Source | Figure |
|---|---|---|---|---|
| pCU-PDC1-GFP | pH078 | CEN, URA3, yeGFP | [71] | |
| CgMsn4-GFP | pH153 | pCU-PDC1-CgMsn4-yeGFP | This study | Figs 5, 6 |
| CgMsn2-GFP | pH152 | pCU-PDC1-CgMsn2-yeGFP | This study | S8 Fig |
| ScDAL80pr-GFP | pH306 | pKW431-*ScDAL80pr*-yeGFP | This study | S12 Fig |

**Table 3. qRT-PCR Primers used in this study.**

| Primer Description | ID | Sequence (5' to 3') | Source | Figure |
|---|---|---|---|---|
| CgCTA1 reverse qPCR primer | oH423 | 5'– CTGTCTTGGTTTGGAATTTGGTA –3' | This study | Fig 3 |
| CgCTA1 forward qPCR primer | oH424 | 5' – ATCCCTGTCAACTGCCCATAC –3' | This study | Fig 3 |
| CgACT1 reverse qPCR primer | oH1347 | 5'– CCACTTTCGTCGTATTCTTGCTTG –3' | This study | Fig 3 |
| CgACT1 forward qPCR primer | oH1348 | 5'– GACCAAACTACTTACAACTCC –3' | This study | Fig 3 |

oxidative stress response genes based on the literature (S1 Table) [33,34]. To identify their homologs in *C. glabrata*, we downloaded the orthology and best-hit mapping between the two species from CGD. The best-hit mapping was based on less stringent criteria and only used when an orthology mapping was not available. The resulting table of gene expression log2 fold changes is available in S2 Table.

## *CTA1* induction by quantitative real-time PCR (qRT-PCR)

*C. glabrata* cells were mock-treated or treated with phosphate starvation or 1.5 mM H$_2$O$_2$ for 45 minutes. ~2-4x10$^7$ cells were harvested by centrifugation, snap-frozen in liquid nitrogen and stored in -80°C until RNA extraction. Two biological replicates were collected per condition. RNA was extracted using a MasterPure Yeast RNA purification kit (Biosearch Technologies, MPY03100). cDNA was synthesized using SuperScript III First-Strand Synthesis System (Invitrogen, 18080051) and treated with RNase A (ThermoFisher, EN0531). We used the LightCycler 480 SYBR Green I Master (Roche, 04707516001) and qRT-PCR was performed on a Roche LC480 instrument. Three technical replicates were performed for each sample. The PCR program was 95°C for 5 minutes, followed by 45 cycles of 95°C for 10 seconds, 60°C for 20 seconds, and 72°C for 20 seconds. Data analysis was done using the instrument software, including automatic baseline subtraction and C$_T$ value quantifications. *ACT1* was used as a reference gene and its C$_T$ values were subtracted from those of *CTA1*. We then subtracted the mean of the $\Delta$C$_T$ for the mock-treated samples from the technical replicates for the phosphate starvation and H$_2$O$_2$-treated ones to obtain $\Delta\Delta$C$_T$, which were used for downstream analyses.

## Cta1-GFP time course by flow cytometry

Cells were grown to mid-log phase in SC medium in 96 deep-well plates (VWR, 82051). After washing with ddH$_2$O twice and spinning down at 3,000g for 5 minutes, cells were resuspended in 0mM Pi SC medium, H$_2$O$_2$-containing SC medium or SC medium (7.5mM Pi), incubated at 30°C shaking at 200 rpm for 4 hrs. At the indicated time points, an aliquot of cell culture (40–50uL) was drawn and diluted into ddH$_2$O (360uL–400uL) for flow cytometry on an Attune NxT instrument fitted with an autosampler (Thermo Fisher). For each sample, 30,000 events were recorded; non-cell events and doublets were removed based on FSC-H and FSC-W. Median fluorescence intensity (MFI) was used for Cta1-GFP protein expression analysis.

## Binding site prediction for *CTA1* promoter

To predict the binding sites (BS) for Skn7 and Msn4 in the *CgCTA1* promoter, we first retrieved the 1000 bp DNA upstream of the start codon for the *CTA1* gene in the *C. glabrata* CBS138 genome (CGD, version s02-m02-r09). We then scanned this sequence with the expert curated motifs for Skn7 and Msn4 from the YeTFaSCo (yetfasco.ccbr.utoronto.ca) database. To perform the scanning, we first selected the two motifs for Skn7 and single expert curated motif for Msn4, "added them to the cart", and used the "scan using cart" function. We adopted

the default 75% minimum percent of maximum score cutoff, and set the background percent A/T to 0.3, which was based on the mean GC% in 1000 bp upstream sequences in the *C. glabrata* genome calculated in-house. The motifs and prediction results were available in S1 Text.

## Fluorescent microscopy for Msn4 and Msn2 nuclear localization

*C. glabrata* cells were transformed with the indicated plasmids and grown on SC Ura- plates for selection. Transformed cells were then grown and maintained in SC Ura- liquid medium prior to the experiments. *C. glabrata* cells were grown to mid-log phase in SC Ura- medium and treated with SC, glucose starvation, or 0 mM Pi SC media for 30 mins. Subsequently, 4',6-diamidino-2-phenylindole (DAPI, Sigma, D9542) was diluted in the respective medium and added to the cultures at a final concentration of 10 ug/ml. Cells were incubated with DAPI in the dark for 15 minutes before DAPI was removed and cells were washed twice with ddH$_2$O. Finally, cells were re-resuspended in the treatment medium. Fluorescent imaging was performed on a Leica SP8 confocal imaging system (Carver Center for Imaging, the University of Iowa). A 488 nm laser was used for yeGFP excitation and 495–585 nm for emission; a 405 nm laser was used for DAPI excitation and 430–550 nm for emission. In *S. cerevisiae*, Msn2-mCherry and Msn4-mCitrine cells were prepared as described above excluding the DAPI staining. For Msn4-mCitrine, 488 nm was used for excitation and 501–547 nm for emission; for mCherry, 552 nm was used for excitation and 570–649 nm for emission; for visualizing the nucleus with the Nhp6a-iRFP marker, a 638 nm laser was used for excitation and 648–790 nm for emission.

## Total and P-Rps6 Western Blot

Mid-log phase cells were subjected to either mock or starvation treatments in parallel. Cell lysis buffer (0.1M NaOH, 0.05M EDTA, 2% SDS, 2% β-mercaptoethanol) and loading buffer (0.25M Tris-HCL pH6.8, 50% Glycerol, 0.05% bromophenol blue) were prepared and protein extraction was performed as described in [78]. Cell lysates were separated by SDS-PAGE at 200 volts for 40 minutes, transferred onto nitrocellulose membranes (Bio-Rad, 1662807), and stained with a total protein stain (Li-COR, 926–11015), which we used for normalization in place of a loading control, as recommended by Li-COR. The membranes were blocked with 3% BSA (RPI, A30075) at room temperature for 1 hr. Two blots for the same samples were probed with either anti-phospho (Ser/Thr) Akt substrate rabbit polyclonal antibody (RRID:AB_330302, Cell Signaling Technology # 9611) or Anti-RPS6 rabbit antibody (RRID:AB_945319, Abcam ab40820) to detect the phosphorylated Rps6 (P-Rps6) and total S6, respectively. After washing, the blots were incubated with a secondary antibody (RRID:AB_621843, LI-COR, 926–32211) and imaged on a Li-COR Odyssey Fc scanner (Carver Center for Genomics, the University of Iowa). The intensity of P-Rps6 and total Rps6 were quantified using the Li-COR Image Studio software, following the manufacturer's instructions for the Revert 700 Total Protein Stain for Western Blot Normalization. Briefly, the band of interest for each lane was selected using same-sized rectangles. We then subtracted the background, calculated as the median of the top and bottom regions of a band. We did the same analysis for the total protein stain image, except we included the entire lane where bands are visible. We then normalized the Western blot signals in each sample to the corresponding total protein stain to obtain the intensity for P-Rps6 or total Rps6. Finally, we divided the two values to get the P-Rps6/total Rps6 ratio.

## Assaying cell morphology by light microscopy

To determine the percent of unbudded cells during a phosphate starvation time course, wild type *C. glabrata* and *S. cerevisiae* cells were grown as described above to mid-log phase,

collected by centrifugation, washed once with pre-warmed 0 mM Pi media and diluted to 5x10$^6$ cells/mL in 0 mM Pi media. At 0, 20, 45, 75, 105 and 135 minutes into the time course, an aliquot of cells was taken from each sample, sonicated for 5s to break up any cell clumps, diluted 10-50x and counted on a hemocytometer (Marienfeld, 0640030) under a light microscope. The number of unbudded cells over the total number of cells were recorded by manually examining > 100 cells over four fields of view. The numbers from the four fields of view were summed for downstream analysis.

## Statistical analyses and tests

For *quantitative ASR assays* using CFU, the data from each replicate experiment was used to calculate survival rates with or without the primary stress, expressed as r = CFU$_{MO}$ / CFU$_{MM}$ and r' = CFU$_{PO}$ / CFU$_{PM}$. The fold change in survival due to the primary stress, expressed as r'/r, aka ASR-score, was calculated for each experiment. We estimated the mean and 95% confidence interval for the ASR-score with 1000 bootstraps using the smean.cl.boot() function in the Hmisc package. To identify equivalent strengths of ROS for different species and strains, a Mann-Whitney U test was used to test for differences in the basal survival rates (r) and a two-sided *P*-value was reported. To assess the statistical significance of the ASR effect in a strain, we used the paired r and r' estimates from each replicate and performed a Wilcoxon signed-rank test if n > = 4. When n < 4, a paired Student's t-test was used since the former test has very low power at such a small sample size. The one-sided *P*-value, with the alternative hypothesis being r' > r, was reported in the legend and text. To compare the ASR effect sizes between two strains, a Mann-Whitney U test (aka Wilcoxson rank-sum test) was performed for the paired data. A two-sided *P*-value was reported. When more than one comparison was made, a Bonferroni correction was applied. For *transcriptomic comparisons*, we performed a t-test for the triplicates of the log2 fold changes for 29 genes in both species. The alternative hypothesis was that the true log2 fold change was greater than 0 (induced under phosphate starvation). The resulting *P*-values were adjusted for multiple comparisons using the Benjamini-Hochberg procedure (FDR). Genes with an FDR < 0.05 and a log2 fold change > 2 were considered significantly induced. For *CTA1 induction by qRT-PCR*, the $\Delta\Delta C_T$ values from the three technical replicates for the phosphate starved, H$_2$O$_2$-treated and mock-treated samples were analyzed using a linear model, with the two stress conditions treated as dummy variables and the biological replicate included as a covariate (R: lm(ddCt ~ treatment + strain), where treatment is a factor with levels = {"Mock", "-Pi", "H$_2$O$_2$"}). Raw *P*-values for each level were reported. For Cta1-GFP basal and induced levels, a linear model was used to compare the different genotypes and the *P*-values were Bonferroni-corrected (multiplied by 5 mutants). For *Msn4 nuclear localization*, cells with nuclear-localized vs cytoplasmic Msn4 were counted under a fluorescent microscope (>100 cells in total per strain x condition). A Fisher's Exact Test (FET) was performed for cells with or without nuclear-localized Msn4 and between species/genotypes. *P*-values were calculated using an exact method. When multiple comparisons were made, a Bonferroni correction was applied unless otherwise noted in the text or legend. For *Western Blot analysis of P-Rps6 / total Rps6*, the samples being compared were run on the same gel for each replicate. Thus, a paired Student's t-test was performed, and a two-sided *P*-values after Bonferroni correction was used to assess the significance of the difference between the treatment conditions.

## Supporting information

**S1 Fig. Basal survival rates and phosphate starvation induced acquired resistance for H$_2$O$_2$ at different primary and secondary stress conditions.** (A) Basal survival rates (r) at different

$H_2O_2$ concentrations in *C. glabrata* and *S. cerevisiae* were quantified using Colony Forming Unit (CFU) ratios between cells treated with $H_2O_2$ and mock treated ones. The red dots and vertical lines show the mean and 95% confidence intervals based on 1000 bootstrap replicates. Individual data points are shown in different shapes grouped by the date of the experiment. (B) Acquired Stress Resistance (ASR) at different $H_2O_2$ secondary stress levels in the two species. ASR-score is defined as the fold increase in survival after the $H_2O_2$ treatment as a result of the primary stress (phosphate starvation). r' is the survival rate with the primary stress and r is the same as in (A), i.e., without the primary stress. The red dots and lines have the same meanings as in (A). (C) ASR for $H_2O_2$ at different primary stress length. 100 mM and 10 mM of $H_2O_2$ were used as the secondary stress for the two species as in Fig 1. The dotted line shows an ASR score of 1 (i.e., no increase in survival due to the primary stress). A paired t-test was performed on the underlying survival rates (r and r') for each of the six species-by-duration combinations. After Bonferroni correction, all three tests in *C. glabrata* had $P < 0.05$, while all three tests in *S. cerevisiae* yielded $P > 0.5$.
(PDF)

**S2 Fig. Phosphate starvation-induced ASR for $H_2O_2$ diverge between species while other primary stresses show similar effects.** ASR experiment was performed as in Fig 1, with the exception of using different primary stresses as indicated on the top: Mock—rich SC medium; HS–Heat shock at 43C; -Glu—0.05% glucose (as opposed to 2% in SC); -Pi—no phosphate SC medium. Images were taken 14 hrs and 21 hrs post spotting for *C. glabrata and S. cerevisiae* respectively.
(PDF)

**S3 Fig. Phosphate starvation's ASR effect depends on the type of ROS.** We tested the ability of phosphate starvation to provide acquired resistance for two additional ROS, i.e., tert-butyl hydroperoxide and menadione sodium bisulfate (MSB), and compared them to $H_2O_2$, each at the indicated concentration. No ASR effect was observed for tBOOH (mean ASR-score = 0.71, 95% CI [0.49, 0.88], $P = 1$). There is moderate ASR for MSB (mean ASR-score = 2.41, 95% CI [1.79, 3.00], raw $P = 0.016$). ASR for $H_2O_2$ is the strongest (mean ASR-score = 16.4, 95% CI [7.7, 25.3], raw $P = 0.031$).
(PDF)

**S4 Fig. *CTA1* complement strain rescues basal $H_2O_2$ survival and ASR defects in *cta1Δ*.** Putting either *CTA1*-GFP (A) or the untagged *CTA1* (B) back to the endogenous locus in the *cta1Δ* background restored the resistance to $H_2O_2$ compared with the wild type strain. Each strain was treated at the indicated concentrations of $H_2O_2$ for 2 hours, then spotted onto YPD plates and incubated at 30˚C for 48 hours (A) and 20hrs (B). (C) ASR in the wild type, *CTA1* complement (*cta1::CTA1*) and *cta1Δ* strains. The experiment was conducted similarly as in Fig 3 for wild type and *cta1Δ* strains. Concentrations of $H_2O_2$ were calibrated to achieve a similar basal survival rates (open circles, Kruskal-Wallis rank sum test for differences among the three groups $P = 0.38$). ASR-scores for the three strains are (with 95% CI and Wilcoxon signed-rank test P-values in the parenthesis): wild type 23.9 ([14.9, 36.5], $P = 0.048$); *cta1::CTA1* 15.4 ([9.5, 21.3], $P = 0.048$); *cta1Δ* 5.1 ([2.0, 10.6], $P = 0.093$). The difference in ASR-score is significant between *cta1Δ* and wild type, but not between *cta1::CTA1* and wild type (Mann-Whitney U test $P = 0.034$ and 0.48, respectively, Bonferroni-corrected).
(PDF)

**S5 Fig. Induction of *CTA1* provides acquired resistance to $H_2O_2$ in *C. glabrata*.** To test if pre-inducing *CTA1* is sufficient to provide ASR for $H_2O_2$, we replaced the endogenous *CTA1* promoter with the promoter of the *C. glabrata MET3* gene. When grown in SC medium

lacking methionine and cysteine (-MC), *CTA1* was induced to a comparable level as in the endogenous *CTA1pr-CTA1* strain under phosphate starvation at 45 minutes (A). Dots represent the mean of at least 3 biological replicates, and the error bars the 95% confidence interval by bootstrapping. The line is the LOESS fit to the data. The endogenous *CTA1pr-CTA1* has a basal expression level that is higher than the *MET3pr-CTA1* (0 min). We also confirmed that the -MC media itself did not provide ASR in the wild type *C. glabrata* cells (B, top). Note that in this set of ASR experiments, all SC medium containing $H_2O_2$ also lacked methionine and cysteine (C). This allows the *MET3pr*-CTA1 to be induced during the secondary stress, mimicking what the wild type strain experiences during the $H_2O_2$ stress. Using this strain, we found that inducing *CTA1* during the primary stress significantly enhanced the survival of *C. glabrata* cells during the secondary oxidative stress, i.e., providing ASR (B, bottom two rows, -MC vs Mock).
(PDF)

**S6 Fig. Cta1 is not required for survival during the primary phosphate starvation, and is not as important for ASR with mild $H_2O_2$ used as the primary stress.** (A) (Left) Deleting *cta1* and the OSR TF, *yap1*, had no defect on survival under phosphate starvation, while deletion of *pho4*, which is responsible for the PHO response, showed severe growth defects. Mid-log phase cells of the indicated genotypes were spotted on no phosphate SC plates, incubated at 30°C for 48 hours. (Right) Same as (Left) but spotted onto SC plates with 7.5 mM Pi. All images are representatives of >3 biological replicates. (B) *CTA1*'s importance for ASR is dependent on the primary stress type. ASR for wild type and *cta1Δ* were tested with various primary stresses, including phosphate starvation (-Pi), glucose starvation (-Glu, 0.02% glucose), mild $H_2O_2$ (1.5 mM $H_2O_2$). ASR experiment was performed as described in the text.
(PDF)

**S7 Fig. TFΔ effects on Cta1-GFP induction under $H_2O_2$ stress.** Cta1-GFP induction under 2 mM $H_2O_2$ for 4 hours. Yap1, Skn7 play a critical role; Msn4 has a minor contribution while Msn2 is not important. Same features as in Fig 4A.
(PDF)

**S8 Fig. *C. glabrata* Msn2 (CgMsn2) and *S. cerevisiae* Msn2 (ScMsn2) translocate into the nucleus upon phosphate starvation.** Cellular localization of CgMsn2 (A) and ScMsn2 (B) under phosphate starvation (-Pi) and no stress conditions. All treatments were for 45 minutes. From left to right: i. CgMsn2-yeGFP and ScMsn2-mCherry (pseudo color); ii. nucleus staining with DAPI in *C. glabrata* or Nh6a-iRFP in *S. cerevisiae*; iii. merge of i and ii; iv. bright field; v. percentage of cells with nuclear-localized CgMsn2 or ScMsn2 under each condition (n = number of cells quantified). Scale bars all represent 5 μm in length. DAPI stain more than the nucleus in live *C. glabrata* cells as observed by others [42].
(PDF)

**S9 Fig. *RIM15* complement strain rescues *rim15Δ* ASR defects.** We restored *RIM15* at the endogenous locus on the *rim15Δ* background and compared its ASR effect to that in the wild type and *rim15Δ* strains. Concentrations of H2O2 were calibrated to achieve a similar basal survival rates (open circles, Kruskal-Wallis rank sum test among the three groups $P = 0.27$). ASR-scores for the three strains are (95% CI and Wilcoxon signed-rank test *P*-values in the parenthesis): wild type 9.5 ([5.3, 16.9], $P = 0.048$); *rim15::RIM15* 11.6 ([8.6, 15.2], $P = 0.048$); *rim15Δ* 5.3 ([3.4, 7.5], $P = 0.048$). Based on Mann-Whitney U test, the ASR-score is not significantly different between either *rim15Δ* and wild type, or *rim15::RIM15* and wild type (Holm-Bonferroni-corrected *P*-values = 0.6 and 0.3, respectively). *rim15Δ* has relatively weak effects on ASR (Fig 6D) and the wild type's survival rates with primary stress has relatively large

variance in this experiment. Both could result in the lack of statistical significance in the comparison between *rim15Δ* and wild type. Instead, the difference in ASR-score between *rim15Δ* and *rim15::RIM15* has a *P*-value of 0.045. Combined, we conclude that *rim15Δ* reduces ASR and the complement strain rescues the defect.
(PDF)

**S10 Fig. TORC1 inhibition by phosphate and nitrogen starvation in diverse yeasts.** (A) Western blot for P-Rps6 (top) and total Rps6 (bottom) in log phase *C. glabrata*, *S. cerevisiae*, *K. lactis* and *L. walti* cells, grown in rich, nitrogen starvation (-N) and phosphate starvation (-Pi) media for 1 hr at 30˚C. This blot is representative of three biological replicates. The subset of the image including *S. cerevisiae* and *C. glabrata* was already shown in Fig 7A. The cladogram on the top depicts the phylogeny between the yeasts; the arrow on the top blot indicates the band for the P-Rps6; the two dotted lines in the bottom blot indicate the bands for total Rps6 used for quantitative analysis. (B) Quantification of the ratio of P-Rps6 to total Rps6 based on three replicates. While the trend was obvious, the small sample size (3) limited the power of a paired Student's t-test used to compare the starvation conditions to the rich condition in each species, which resulted in *P*-values that were not significant at a 0.05 level after Bonferroni correction.
(PDF)

**S11 Fig. Sch9-3E phosphomimetic mutant in *C. glabrata*.** (A) Schematic alignment of Sch9 protein in *S. cerevisiae* and *C. glabrata*. The three TORC1 targeted S/T sites in ScSch9, based on Urban *et al*. 2007, and the corresponding sites in the orthologous CgSch9 based on the alignment were labeled with a triangle and the mutations were labeled on the top. The region containing the three S/T sites were shown below as a pairwise sequence alignment below. The S/T sites were shown in red bold fonts. For the second site, S758 (in ScSch9), the corresponding site in CgSch9 are flanked by two serines, making it difficult to determine whether and which serine may be the authentic phosphorylation site. We therefore decided to make three mutations to turn the site into the same sequence as in the ScSCH9-3E mutant. (B) To replace the endogenous *SCH9* gene in *C. glabrata* with the mutant alleles, we added an antibiotic NAT marker at the end of either the wild type *CgSCH9* gene or the *CgSCH9-3E* allele and performed allele swaps. The first construct provides a control for the effect of disrupting the 3' UTR by adding the NAT marker.
(PDF)

**S12 Fig. *DAL80* induction kinetics under nitrogen and phosphate starvation in *S.cerevisiae* and *C. glabrata*.** *ScDAL80pr*-GFP expression levels were monitored via flow cytometry over a time course of 4 hours during nitrogen (-N) or phosphate starvation (-Pi) and rich medium conditions. Dots represent the mean of Median Fluorescent Intensity from > 3 biological replicates; the error bars represent the 95% CI based on 1000 bootstraps. The lines are LOESS fit to the means. The top row shows all three conditions while the bottom row shows the same data without -N to better visualize the induction under -Pi.
(PDF)

**S13 Fig. Phosphate starvation response is partially induced in *C. glabrata* after the yeast is engulfed by human macrophages.** Shown are the phosphate homeostasis genes induced during phosphate starvation in *C. glabrata*. Gene list is from [25]. PolII ChIP-seq time course data are from [31], sampled at 0.5h, 2h, 4h, 6h and 8h post infection. For each gene, the ChIP-seq RPKM values were divided by the mean such that the normalized values represent fold changes over the mean for that gene. Systematic gene IDs were labeled on the top of each panel. Gene names were shown inside the panel (names in parentheses were based on *S.*

*cerevisiae* homologs). Colors represented gene functional categories as explained below.
(PDF)

**S14 Fig. Genes induced after being engulfed by macrophages are also induced by a short-term phosphate starvation in *C. glabrata*.** Data and plots are similar to Fig 2C–2F. Gene sets were based on [31] (S2 and S3 Figs for Fig 1 therein). Gene names were based on *S. cerevisiae*. An asterisk meant the gene was significantly induced in the species at an FDR of 0.05.
(PDF)

**S15 Fig. Percent unbudded cells during phosphate starvation time course.** Cells were taken at 0, 20, 45, 75, 105 and 135 minutes during the time course and assayed for cell morphology by light microscopy (Materials and Methods). The black and dark red dots and lines represent two biological replicates. The dots are the percentage of unbudded cells from >100 total cells examined. The error bars are the 95% confidence interval for the binomial proportion based on normal approximation (Wald Interval).
(PDF)

**S1 Text. Predicting TF binding sites in *C. glabrata CTA1* promoter.** Supplementary method.
(DOCX)

**S1 Table. *S. cerevisiae* OSR genes.** Gene list curated from the literature in *S. cerevisiae* in response to $H_2O_2$.
(XLSX)

**S2 Table. Log2 fold changes in *S. cerevisiae* and *C. glabrata* after 1 hour of phosphate starvation.** Average log2 fold changes relative to non-starved cells from 3 biological replicates were shown for both species, for the genes listed in S1 Table.
(XLSX)

## Acknowledgments

We would like to thank Lan Qing, Nandan Rai and Koon Ho Wong for kindly providing the PolII ChIP-seq data for phagocytosed *C. glabrata*. We would like to thank Dr. Nan Hao for sending us the *S. cerevisiae* Msn2 and Msn4 fluorescently tagged strains. We thank Dr. Anthony Pannullo and other members of the Gene Regulatory Lab members for reading and giving feedback on the manuscript. We also want to thank Drs. Daniel Summers, Damian Krysan, Josep Comeron and Andrew Capaldi for giving suggestions on the work. Drs. Jan Fassler, Veena Prahlad and Bing Luan read an early draft of the paper and gave valuable feedback.

## Author Contributions

**Conceptualization:** Jinye Liang, Bin Z. He.

**Data curation:** Jinye Liang, Hanxi Tang, Lindsey F. Snyder, Christopher E. Youngstrom, Bin Z. He.

**Formal analysis:** Jinye Liang, Hanxi Tang, Lindsey F. Snyder, Christopher E. Youngstrom, Bin Z. He.

**Funding acquisition:** Bin Z. He.

**Investigation:** Jinye Liang, Hanxi Tang, Lindsey F. Snyder, Christopher E. Youngstrom, Bin Z. He.

**Methodology:** Jinye Liang, Hanxi Tang, Lindsey F. Snyder, Christopher E. Youngstrom, Bin Z. He.

**Project administration:** Bin Z. He.

**Resources:** Jinye Liang, Bin Z. He.

**Supervision:** Bin Z. He.

**Validation:** Jinye Liang, Bin Z. He.

**Visualization:** Jinye Liang, Bin Z. He.

**Writing – original draft:** Jinye Liang, Bin Z. He.

**Writing – review & editing:** Jinye Liang, Hanxi Tang, Lindsey F. Snyder, Christopher E. Youngstrom, Bin Z. He.

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
