## [Decision Letter · Decision Letter 0]

10 Jul 2023

Dear Assistant Professor He,

Thank you very much for submitting your manuscript "Divergence of TORC1-mediated Stress Response Leads to Novel Acquired Stress Resistance in a Pathogenic Yeast" for consideration at PLOS Pathogens. As with all papers reviewed by the journal, your manuscript was reviewed by members of the editorial board and by several independent reviewers. In light of the reviews (below this email), we would like to invite the resubmission of a significantly-revised version that takes into account the reviewers' comments.

All three reviewers agree that the manuscript is important but requires further revisions.

The most important set of comments pertains to validation of some of your data, including signal strength and imaging of the blots, absence of complement strain and determining the sufficiency of CTA1, and the use of only one type of oxidative stressor.

Refining these issues will be important to solidify the important findings of this manuscript.

Also, please make an effort to shorten the manuscript.

This can be achieved by removing repetitive statements and focusing on the most important findings while limiting discussion outside of these major areas of focus.

We cannot make any decision about publication until we have seen the revised manuscript and your response to the reviewers' comments. Your revised manuscript is also likely to be sent to reviewers for further evaluation.

Sincerely,

Michal A Olszewski, DVM, PhD

Section Editor

PLOS Pathogens

Michal Olszewski

Section Editor

PLOS Pathogens

Kasturi Haldar

Editor-in-Chief

PLOS Pathogens

orcid.org/0000-0001-5065-158X

Michael Malim

Editor-in-Chief

PLOS Pathogens

orcid.org/0000-0002-7699-2064

All three reviewers agree that the manuscript is important but requires further revisions.

The most important set of comments pertains to validation of some of your data, including signal strength and imaging of the blots, absence of complement strain and determining the sufficiency of CTA1, and the use of only one type of oxidative stressor.

Refining these issues will be important to solidify the important findings of this manuscript.

Also, please make an effort to shorten the manuscript.

This can be achieved by removing repetitive statements and focusing on the most important findings while limiting discussion outside of these major areas of focus.

Reviewer's Responses to Questions

**Part I - Summary**

Reviewer #1: In this paper, the authors investigate the role of acquired resistance to oxidative stress following phosphate starvation in the pathogenic yeast C. glabrata and its non-pathogenic relative S. cerevisiae. The approach is rigorous, with a logical flow identifying an effector responsible for the observed phenotype, as well as a systematic approach to identifying the signaling pathway and transcription factors responsible. The significance is high, in that it also demonstrates that conserved factors between the model yeast and related pathogen, there has been a re-wiring that leads to activation of an oxidative stress response in response to phosphate starvation only in the pathogen. They find that phosphate starvation leads to ASR in C. glabrata, but not S. cerevisiae, and that this is primarily due to upregulation of the catalase, Cta1. The authors then elaborate on the mechanism of Cta1 upregulation, which they show is partially dependent on the transcription factors Skn7 and Msn4, the latter of which is regulated by the TOR pathway under phosphate starvation. Overall, these results are convincing, and the authors' conclusions are well-supported by the data. Most initial concerns were addressed by the supplementary data, leaving only a few minor concerns.

Reviewer #2: The study by Liang et. al. demonstrates that phosphate starvation induces a strong acquired resistance to H2O2 stress in Candida glabrata but not in Saccharomyces cerevisiae. This species-specific response is accompanied by the significant induction of oxidative stress-related genes in C. glabrata. The researchers further identify the CgCTA1 gene as essential for the acquired resistance to severe H2O2 stress during phosphate starvation, and they uncover the joint contribution of transcription factors CgMsn4 and CgSkn7 in inducing CgCTA1 expression. Additionally, the study reveals that CgMsn4 translocates to the nucleus upon phosphate starvation in C. glabrata, in contrast to its ortholog in S. cerevisiae. The involvement of Rim15, a Greatwall kinase homolog, and the inhibition of TORC1 signaling by phosphate starvation are also highlighted as mediators of the acquired stress resistance in C. glabrata. This study demonstrates a strong and well-controlled experimental design, yielding robust results. The data analysis appears thorough and appropriately conducted.

Reviewer #3: In this manuscript the authors use a number of different approaches to attempt to tease apart the mechanisms by which stress response networks are wired in the human fungal pathogen Candida glabrata and how this is evolutionarily different from that of S. cerevisiae.

They have focused on the link between phosphate starvation inducing oxidative stress response genes – cross-tolerance or as they refer to it as acquired stress resistance.

The authors have identified TORC1 being differentially inhibited by phosphate starvation when comparing C. glabrata and S. cerevisiae, as one might expect give the C. glabrata cells have to overcome/evade immune cells and phagocytosis.

Overall this is a extremely well written manuscript, with all of the big hitting references on the topic cited, however it is in its current version, too long, from reading it numerous times now, I strongly recommend that it be split into two more cohesive stories and the comparison element with S. cerevisiae is removed as this does not add to the story – I know that within this field S. cerevisiae is often used as a comparator due to being more closely related to C. glabrata than say C. albicans but in this case it is distracting from the concepts of ASR and cellular rewiring occurring the C. glabrata.

The figures for the manuscript are of a very high standard but there is an overuse of schematics (F1, F2, F4, F6, F7, F8) a better way to present may be in one final figure describing the overall model presented herein. In addition, I would like to see some of the data from SF1 moved into the main body of the text.

It would also be very interesting to see if the phosphate starvation also offered protection against other oxidative stressors such as tBOOH and how that effect the up/down regulation of Msn2/4 and Sch9.

The general execution of the manucript is of a high standard.

**Part II – Major Issues: Key Experiments Required for Acceptance**

Reviewer #1: No Major Issues

Reviewer #2: 1. One concern regarding the study is the lack of clarity on whether the induction of CTA1 alone is sufficient to confer acquired stress resistance (ASR) to H2O2. While the study indicates that CgCTA1 induction is required for ASR during phosphate starvation, it does not directly address whether CTA1 induction alone is adequate to establish the resistance phenotype. Further investigations, such as genetic manipulation experiments, are needed to determine the sufficiency of CTA1 induction in conferring ASR for H2O2.

2. Another concern relates to the functional characterization of CgMsn4 and CgSkn7 in the context of acquired resistance to H2O2 stress in C. glabrata. While the study suggests that these transcription factors contribute to CTA1 induction during phosphate starvation, it would be nice to show if CgMsn4 and CgSkn7 knock out strains display defects in the acquired resistance to H2O2 stress.

3. Finally, the reviewer’s concern is the lack of complemented strain. The inclusion of a few key complemented strains (such as CTA1) would strengthen the conclusions drawn from the knockout experiments. Additionally, it would be beneficial to mention the background of the parental strain used in the study.

Reviewer #3: Screening on addtional oxidative stress chemicals post phophate starvation - is this limited to hydrogen peroxide only?

**Part III – Minor Issues: Editorial and Data Presentation Modifications**

Reviewer #1: • Although the conclusions drawn from panel 7A remain convincing, the non-phospho RPS6 antibody does not give a clear signal, and thus it's unclear whether quantitation (Fig. 7B) is suitable. Were both major bands treated as RPS6 during quantification?

• In supplementary figure 3, both panels, it appears that two different plate images are juxtaposed without space between them. This is misleading. There should be white space between the images from separate plates.

• It’s odd that there is a figure callout to S1 in the introduction. Is there a way that this can be incorporated into the results section?

Reviewer #2: (No Response)

Reviewer #3: The figures are overly populated with schematics - one summary figure would be beneficial.

PLOS authors have the option to publish the peer review history of their article (what does this mean?). If published, this will include your full peer review and any attached files.

Reviewer #1: No

Reviewer #2: No

Reviewer #3: No
---

## [Decision Letter · Decision Letter 1]

11 Oct 2023

Dear Assistant Professor He,

We are pleased to inform you that your manuscript 'Divergence of TORC1-mediated Stress Response Leads to Novel Acquired Stress Resistance in a Pathogenic Yeast' has been provisionally accepted for publication in PLOS Pathogens.

Best regards,

Michal A Olszewski, DVM, PhD

Section Editor

PLOS Pathogens

Michal Olszewski

Section Editor

PLOS Pathogens

Kasturi Haldar

Editor-in-Chief

PLOS Pathogens

orcid.org/0000-0001-5065-158X

Michael Malim

Editor-in-Chief

PLOS Pathogens

orcid.org/0000-0002-7699-2064

Reviewer Comments (if any, and for reference):

Reviewer's Responses to Questions

**Part I - Summary**

Reviewer #1: No weaknesses. All concerns of this reviewer were addressed.

Reviewer #2: All comments and concerns have been addressed thoroughly. Excellent job!

Reviewer #3: (No Response)

**Part II – Major Issues: Key Experiments Required for Acceptance**

Reviewer #1: None noted.

Reviewer #2: All comments and concerns have been addressed

Reviewer #3: The authors have addressed my comments to the best of their ability in the revised manuscript.

**Part III – Minor Issues: Editorial and Data Presentation Modifications**

Reviewer #1: (No Response)

Reviewer #2: (No Response)

Reviewer #3: (No Response)

PLOS authors have the option to publish the peer review history of their article (what does this mean?). If published, this will include your full peer review and any attached files.

Reviewer #1: No

Reviewer #2: No

Reviewer #3: No

---

## [Editor Report · Acceptance letter]

17 Oct 2023

Dear Assistant Professor He,

We are delighted to inform you that your manuscript, "Divergence of TORC1-mediated Stress Response Leads to Novel Acquired Stress Resistance in a Pathogenic Yeast," has been formally accepted for publication in PLOS Pathogens.

Best regards,

Kasturi Haldar

Editor-in-Chief

PLOS Pathogens

orcid.org/0000-0001-5065-158X

Michael Malim

Editor-in-Chief

PLOS Pathogens

orcid.org/0000-0002-7699-2064